# LONG-HORIZON REASONING AGENT FOR OLYMPIAD-LEVEL MATHEMATICAL PROBLEM SOLVING

## ABSTRACT

Large Reasoning Models (LRMs) have expanded the mathematical reasoning frontier through Chain-of-Thought (CoT) techniques and Reinforcement Learning with Verifiable Rewards (RLVR), capable of solving AIME-level problems. However, the performance of LRMs is heavily dependent on the extended reasoning context length. For solving ultra-hard problems like those in the International Mathematical Olympiad (IMO), the required reasoning complexity surpasses the space that an LRM can explore in a single round. Previous works attempt to extend the reasoning context of LRMs but remain prompt-based and built upon proprietary models, lacking systematic structures and training pipelines. Therefore, this paper introduces Intern-S1-MO, a long-horizon math agent that conducts multi-round hierarchical reasoning, composed of an LRM-based multi-agent system including reasoning, summary, and verification. By maintaining a compact memory in the form of lemmas, Intern-S1-MO can more freely explore the lemma-rich reasoning spaces in multiple reasoning stages, thereby breaking through the context constraints for IMO-level math problems. Furthermore, we propose OREAL-H, an RL framework for training the LRM using the online explored trajectories to simultaneously bootstrap the reasoning ability of LRM and elevate the overall performance of Intern-S1-MO. Experiments show that Intern-S1-MO can obtain 26 out of 35 points on the non-geometry problems of IMO2025, matching the performance of silver medalists. In addition, it also surpasses the current SOTA LRM on inference benchmarks such as HMMT2025, AIME2025, and CNMO2025.

## 1 INTRODUCTION

Reasoning is a highly intellectual human activity that requires the integration of deductive logic, pattern recognition, and creative problem decomposition to address complex challenges, which is regarded as a significant milestone towards Artificial General Intelligence (AGI) (Sun et al., 2025). In recent years, large reasoning models (LRMs) have made substantial progress in mathematical reasoning, driven primarily by techniques such as Chain-of-Thought (CoT) (Zhang et al., 2022; Wang et al., 2023) and Reinforcement Learning from Verifiable Rewards (RLVR) (Shao et al., 2024; Yue et al., 2025; Zeng et al., 2025). Along with the increasing reasoning capabilities of LRMs, a clear trend is that LRMs are being allocated more thinking budgets for more difficult problems to support the exploration of larger solution spaces and the trial-and-error processes (Zhou et al., 2022; Aggarwal & Welleck, 2025).

However, hardware and data limitations have made unlimited scaling of context length infeasible. Currently, state-of-the-art (SOTA) reasoning models typically support a maximum context length of only 64k or 128k tokens (Yang et al., 2025; Bai et al., 2025; DeepMind, 2025a), insufficient for ultra-challenging problems such as those in International Mathematical Olympiads (IMO) [1]. Figure 1(a) illustrates the logarithmic growth of the required context length with increasing difficulty of the problem, highlighting the mismatch between the existing capacity limits and practical demands. While resource investment can marginally raise this context ceiling, developing a cost-effective paradigm to meet context requirements is more compelling (Li et al., 2025a; Ke et al., 2025).

Some studies have explored multi-round interaction (Motwani et al., 2024) or parallel decoding (Zhang et al., 2024a) to perform long logical deduction in mathematical reasoning. Furthermore,

---

[1] https://imo2025.au

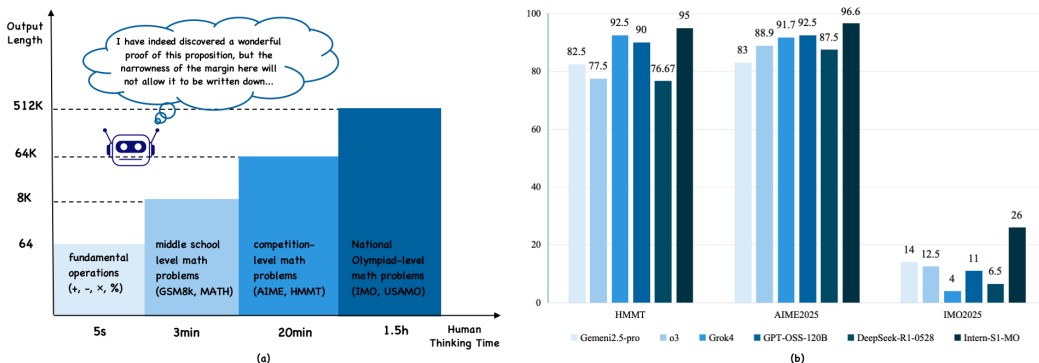

Figure 1: **The motivation (a) and performance(b) of Intern-S1-MO.** As problem difficulty increases, both the average human thinking time and the model token consumption per problem grow exponentially (a), already reaching concerning limits under current development trends. Intern-S1-MO enables LRMs to use about 512K tokens to solve a single problem, achieving state-of-the-art performance on challenging mathematical benchmarks (b).

Huang & Yang (2025) introduced self-reflective with prompt engineering, allowing models to identify flaws in intermediate reasoning steps and refine the outputs. Nevertheless, these approaches still confine problem-solving to a single reasoning cycle (even with internal iterations) rather than building cumulatively upon prior reasoning trajectories, which limits their capacity to leverage historical explorations for further in-depth deduction (Wang et al., 2025). Alternatively, formal language–based search (Ren et al., 2025; Chen et al., 2025; Zhou et al., 2025) shows some promise: by maintaining a structured repository to store and reuse intermediate results, they reduce reliance on model context length. However, the proof verification and state traversal demand extensive iterations, leading to high computational and search overhead. Moreover, formal systems require translating informal descriptions into formal logic, introducing additional costs and hindering the interaction between AI and humans.

Proprietary LRMs (OpenAI, 2025c; DeepMind, 2025b) have reported impressive results on the International Mathematical Olympiad 2025 (IMO2025) problems, yet the research community lacks access to their methodologies and models. In this work, we present Intern-S1-MO, a math reasoning agent framework unconstrained by context length, and solves complex reasoning problems through hierarchical decomposition, a strategy that closely aligns with human problem-solving patterns. Intern-S1-MO achieves unlimited exploration capability through lemma memory management. Specifically, after each single-round reasoning, the agent compresses its current reasoning history into concise sublemmas with a structured memory repository, which enables the agent to recover historical exploration outcomes in subsequent steps. We furthermore design process verification and revision mechanisms to certify the quality of the lemma repository. Notably, Intern-S1-MO enables adaptive control of its reasoning budget: it initiates multi-round exploration only for challenging tasks, ensuring efficient resource allocation.

To support the bootstrapping and online improvement of Intern-S1-MO, we additionally introduce the OREAL-H framework, enabling the agent to enhance its performance on complex problems with online reinforcement learning (RL). Starting from the basic formulation of Outcome Reward Reinforcement Learning (OREAL) (Lyu et al., 2025), OREAL-H exploits the additional reward signal produced by the outcome process verifier (OPV) that is continuous and accelerates training, and is modified for the Hierarchical Markov Decision Process (MDP) formulation to suit the multi-agent setting of Intern-S1-MO.

As a result, Intern-S1-MO establishes new state-of-the-art results across multiple mathematical reasoning benchmarks. As shown in Figure 1(b), on AIME2025 and HMMT2025, it achieves a 96.6% and 95% average pass@1 score, respectively. On the 5 non-geometry problems of International Mathematical Olympiads 2025 (IMO2025), Intern-S1-MO could obtain 26 out of 35 scores, surpassing the silver medalist-level performance (21) of humans. Additionally, we test Intern-S1-MO on CNMO2025, a new benchmark comprising 14 high-school math competition problems (excluding geometry problems) from the recently concluded China National Mathematics Olympiad 2025, on which Intern-S1-MO scores 232.4 out of 260 points. To facilitate rapid reproduction of our agent

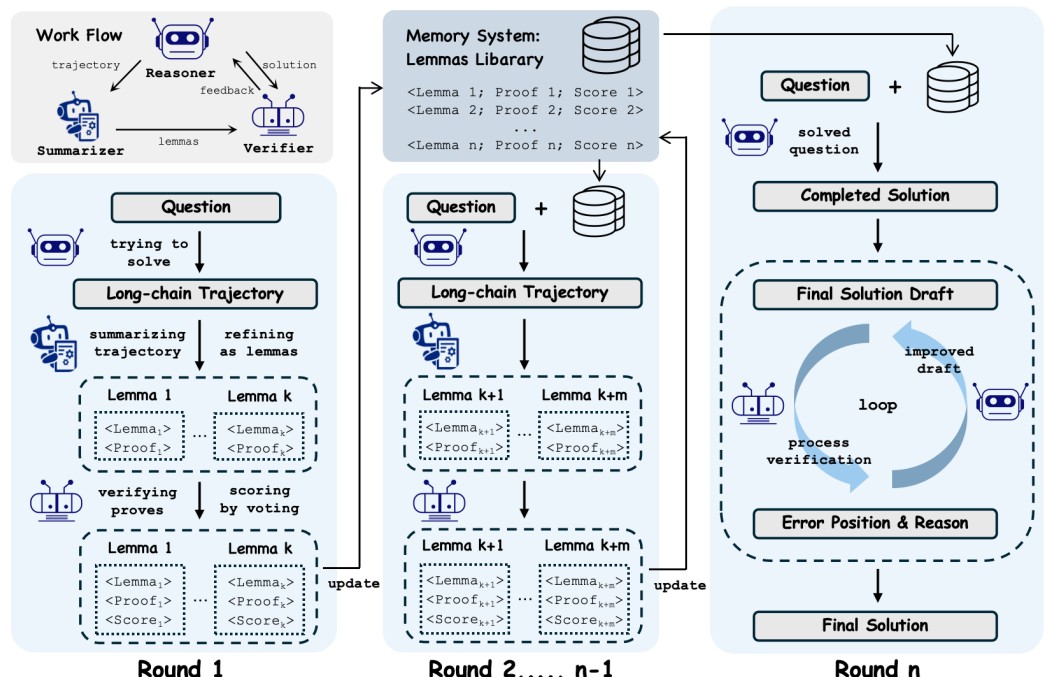

Figure 2: **The agentic framwork of Intern-S1-MO.** In each reasoning round, the reasoner agent tries to solve the question, and the summarizer agent compresses the current reasoning history into a series of lemmas, which will be added to the memory system after being verified by the verifier agent. Except for the first round, the lemma library will be input into the reasoning agent along with the question. In the final round, the solution generated by the reasoner agent undergoes a modification loop, which improves the quality of the solution based on feedback from the verifier agent, until the verification is passed or the maximum number of loop rounds is reached.

framework, we will open-source Intern-S1-mini-MO, a fine-tuned 8B model based on Intern-S1-mini (Bai et al., 2025), specifically optimised for the entire multi-agent system. When integrated with Intern-S1-MO, Intern-S1-mini-MO outperforms models at greater parameter scales, achieving a 90% pass@1 score on AIME2025 and completely solving 3 out of 5 non-geometric problems in IMO2025. Overall, our contributions are as follows:

- We explore multi-round complex reasoning scenarios and propose a multi-agent system, Intern-S1-MO, which effectively extends the reasoning depth of current LRMs by the lemma-based memory management.

- We contribute an RL framework, termed OREAL-H, for optimizing the multi-round performance of Intern-S1-MO on high-difficulty mathematical problems.

- Experiments prove that our Intern-S1-MO outperforms current advanced LRM like Gemini 2.5 Pro. Specifically, Intern-S1-MO can match the performance of silver medalists in IMO2025 and get SOTA in benchmarks like AIME2025, HMMT2025, and CNMO2025.

## 2 BUILDING HIERARCHICAL REASONING AGENTS

To extend the exploration of reasoning, we designed a hierarchical mathematical reasoning agent tailored for complex competition-level mathematical problems, as shown in Figure 2. By enabling recursive subproblem solving, it specifically addresses the aforementioned reasoning limitations constrained by context length. We give a case example in Appendix. F.

**Decomposing Sub-Problems for Lemma Search** Decomposing complex problems into manageable sub-lemmas is a defining feature of human problem-solving for high-difficulty mathematics, as it breaks long-chain logical reasoning into incremental steps. We first observe that state-of-the-art

models already exhibit a degree of reasonable decomposition capability for mathematical problems, though this ability is often undermined by a premature conclusion bias: when reasoning budgets are exhausted, models tend to rush toward incomplete or incorrect final answers instead of acknowledging partial progress. To mitigate this, we refine the model via prompt engineering and targeted training, explicitly enabling it to produce partial deductive progress in single-turn attempts (e.g., deriving intermediate sub-lemmas without forcing a full problem solution). This adjustment aligns the model's behavior with human iterative reasoning and lays the groundwork for cumulative exploration, the complete style requirements are presented in the Appendix A.

**Summarizing Exploration for Memory Maintenance**   The model's reasoning processes for complex problems often include redundant exploratory efforts and trial-and-error content. While this content aids in generating intermediate conclusions, it adds little value to subsequent deductive steps. Such facts enable us to extract only the essential components that drive progress, specifically, validated intermediate lemmas from each reasoning turn and store them in a structured lemma library. This library encourages the agent to reuse historical conclusions during new exploration rounds, allowing for deeper deductions based on prior lemmas rather than reprocessing redundant information. Notably, summarizing compelling exploration is as complex as the exploration process itself, as it requires distilling and checking the logical validity independently. Therefore, we allocate a dedicated reasoning turn after each exploration step to update the lemma library. This computational cost is necessary to ensure the library remains useful for long-chain reasoning.

**Verifying Theorems to Mitigate Error Propagation**   Advanced reasoning models can self-reflect, but if they rely on erroneous historical premises, they will expend significant resources trying to validate questionable results. Such a problem is compounded by error propagation, so that a flawed intermediate conclusion can mislead subsequent deductive directions, leading to circular reasoning or invalid proofs. Fortunately, the verification of lemmas is comparatively more tractable than that of the complete problem. We address this by integrating a theorem verifier that uses parallel sampling to compute confidence scores for each lemma. Specifically, for each lemma, we make the theorem verifier perform n parallel verifications, and the proportion of those correctly identified is used as the confidence score. We believe this improves the reliability of theorem verification, avoiding some false positives or false negatives.

**Verifying Process for Final Proof Completion**   Verifying the validity of final solutions is crucial for obtaining reliable performance feedback, both in evaluation scenarios and reinforcement learning loops. In practice, the verifier serves two main functions: (1) enhancing robustness through test time scaling by aggregating verification results across multiple runs, and (2) providing high-quality feedback signals for iterative revision and reinforcement learning training to further optimize the agent's reasoning precision.

## 3   RL TRAINING FOR EVOLUTION OF MATH AGENTS

### 3.1   PRELIMINARIES

We model the agentic mathematical reasoning process as a *Hierarchical Markov Decision Process*, denoted $\mathcal{M} = \langle \mathcal{S}, \mathcal{U}, \mathcal{V}, r, R, \gamma \rangle$, where $\mathcal{S}$ is the state space (problem context + reasoning trace + verification feedback), $\mathcal{U}$ the high-level meta-action space (e.g., "extract lemmas", "invoke verification", "commit answer"), and $\mathcal{V}$ the low-level token vocabulary. The agent alternates between high-level decisions and low-level generation: at each round $t$, it executes a reasoning action $u_t$ with token sequence $\boldsymbol{v}_t = (v_{t,1}, \ldots, v_{t,T_t}) \sim \pi_\theta^L(\cdot|s_t)$ to produce a reasoning segment. This output is summarized and verified by an external module, yielding natural language feedback which induces an intermediate proxy reward $r_t \in \mathbb{R}$. Upon termination after several rounds, a sparse final reward $R$ indicates correctness of the solution. The training objective is to maximise expected final reward:

$$J(\theta, \phi) = \mathbb{E}_{\pi_\phi^H, \pi_\theta^L}[R].$$

(1)

Leveraging the conditional structure of the hierarchical policy, the per-round advantage can be estimated via a high-level critic $V(s_t)$, updated to satisfy:

$$V(s_t) \leftarrow \mathbb{E}\left[r_t + \gamma V(s_{t+1})\right], \tag{2}$$

where $s_{t+1}$ is the state after applying $u_t$. The advantage for round $t$ is then $A_t = r_t + \gamma V(s_{t+1}) - V(s_t)$. On low-level, we can then perform an online policy gradient conditioned on this advantage, aggregating token-level log-likelihoods within the round:

$$\nabla_\theta J = \mathbb{E}\left[\sum_{t=1}^{K} A_t \cdot \sum_{\tau=1}^{T_t} \nabla_\theta \log \pi_\theta^L(v_{t,\tau} \mid s_t, v_{t,<\tau})\right], \tag{3}$$

**Reward Function** As mentioned in Section 2, we employ a Process Verifier (PV) to assess the logical rigor of complex mathematical proofs. Specifically, the PV examines the agent's final solution and outputs natural language feedback identifying the indices of steps containing logical fallacies. We estimate the PV's confidence via a multi-round voting mechanism. In particular, for problems amenable to outcome supervision, the final reward $R$ is set to 0 if the final answer is incorrect. We further discuss the role of these supervision signals for RL steps in Section 3.3.

### 3.2 CLONING SUCCESS TRAJECTORY FOR COLD START

To prime the agent's adherence to structured reasoning formats and internalise the iterative agentic workflow, we initialize policies via behavioural cloning on filtered trajectories — retaining only rounds $t$ where the output admits a well-formed lemma summary (e.g., syntactically valid, non-empty, logically segmented). Let $\mathcal{D}_{\text{init}} = \{(s_t, \boldsymbol{v}_t)\}$ denote such transitions. The token-level pretraining objective is:

$$\mathcal{L}_{\text{RFT}}(\theta) = -\mathbb{E}_{(s_t, \boldsymbol{v}_t) \sim \mathcal{D}_{\text{init}}}\left[\sum_{\tau=1}^{T_t} \log \pi_\theta^L(v_{t,\tau} \mid s_t, v_{t,<\tau})\right]. \tag{4}$$

Notably, we continuously augment $\mathcal{D}_{\text{init}}$ with question-answer pairs that are filtered by outcome-based scoring, without previous thinking. We observe that the model exhibits emergent generalization: patterns learned from these simplified trajectories boost agentic solving of the same problems, thereby improving the efficiency of positive trajectory discovery during online RL.

### 3.3 OREAL WITH CONJUGATE REWARD UNDER PROCESS JUDGEMENT

We adopt the reinforcement learning framework of Oreal for policy optimization, and introduce two critical adaptations tailored to our Hierarchical MDP setting: (1) credit assignment across high-level reasoning actions is non-trivial due to delayed rewards; (2) the Process Verifier (PV) introduces a continuous, noisy reward signal that deviates from the binary outcome supervision assumed in RLVR setting.

**Progress-Conditioned Advantage via Lemma Dependency Graphs** Existing RLVR training predominantly targets outcome verification (e.g., final answer correctness), which proves insufficient for complex mathematical tasks requiring high process supervision. To align optimization with granular reasoning fidelity, we assign sparse reward signals across reasoning rounds, akin to performing round-level temporal differencing to minimize advantage estimation variance.

To rigorously quantify intermediate progress, we introduce a lemma dependency graph by aggregating reasoning states across multiple rollouts of the same problem. This graph structure captures the probabilistic contribution of specific lemmas to the final proof. Such mechanism functions as a computationally efficient surrogate to Monte Carlo Tree Search (MCTS), providing high-quality value estimation without the prohibitive overhead of extensive search. An example of the lemma graph is presented in Appendix. E

Within this topology, the value of a specific lemma $l$ is not isolated but structurally coupled with the proof's progression. We define the value of a lemma recursively as the expected value of its

subsequent derived lemmas, effectively backpropagating the success probability from the final answer to intermediate steps:

$$v(l) = \mathbb{E}_{l' \in \text{Succ}(l)} \left[ v(l') \right], \tag{5}$$

where $\text{Succ}(l)$ denotes the set of valid lemmas derived directly from $l$ in the dependency graph. For the policy optimization, we anchor credit to rounds that yield verifiable advances. Specifically, for a reasoning round $t$ that generates a set of candidate lemmas $\mathcal{L}_t$, we adopt an optimistic value estimation strategy. We define the state value of round $t$ as the maximum value among its generated candidates, $V(s_t) = \max_{l \in \mathcal{L}_t} v(l)$. The round-level advantage is then computed via the temporal difference error between the best potential of the current round and the next:

$$A_t = r_t + \gamma \max_{l' \in \mathcal{L}t+1} v(l') - \max l \in \mathcal{L}_t v(l). \tag{6}$$

where $r_t$ represents the immediate step reward (e.g., syntactic validity or solving a sub-goal) and $\gamma$ is the discount factor.

For intermediate rounds yielding no new lemmas ($C_t = 0$), the advantage is effectively masked. These formulation ensures that the gradient estimation is driven by the most promising reasoning path discovered at each step, decoupling optimization intensity from trajectory length and effectively filtering out noise from suboptimal branches.

**Conjugate Reward Modeling for Noisy Process Verification**    Process Verification (PV) offers valuable insight into the internal logical consistency of a generated solution by subjecting its intermediate steps to multiple stochastic checks. However, unlike final-answer correctness—which is deterministic—PV feedback is inherently noisy: a solution passing $k$ out of $n$ verification rounds does not guarantee superior reasoning quality, as passes may arise from lucky sampling or superficial plausibility rather than deep correctness. Directly using the empirical ratio $k/n$ as a reward signal risks amplifying this noise, leading to unstable or misguided policy updates that overfit to verification artifacts rather than genuine mathematical rigor.

To address this, we adopt a Bayesian perspective and model the latent reasoning quality $p \in [0, 1]$ as a random variable. We place a uniform prior $p \sim \text{Beta}(1, 1)$, encoding no initial assumption about solution validity. After observing $k$ successful verifications in $n$ independent PV trials, the conjugate Beta-Bernoulli update yields the posterior:

$$p \mid (k, n) \sim \text{Beta}(k + 1, n - k + 1). \tag{7}$$

Instead of using point estimates (e.g., posterior mean), we define the reward as the probability that this solution is *strictly better* than a canonical "completely invalid" baseline—one that fails all $n$ checks ($k = 0$). Let $p_1 \sim \text{Beta}(k + 1, n - k + 1)$ represents the quality of the current solution and $p_0 \sim \text{Beta}(1, n + 1)$ that of the baseline. The reward is then:

$$R(k, n) = \mathbb{P}(p_1 > p_0) = \int_0^1 \int_0^1 \mathbb{I}(p_1 > p_0) \cdot f_{\text{Beta}(k+1,n-k+1)}(p_1) \cdot f_{\text{Beta}(1,n+1)}(p_0) \, dp_1 dp_0. \tag{8}$$

This formulation provides a principled, probabilistically calibrated reward that accounts for uncertainty in the verification process. It naturally suppresses spurious signals from low-pass outcomes while preserving strong gradients for high-confidence valid solutions.

In practice, we fix $n = 4$, balancing verification cost and signal fidelity. Under this setting, $R(4, 4) \approx 5.5$, corresponding to a 99.5% dominance probability over the $R(0, 4) = 0$ baseline, with smoothly interpolated rewards for intermediate cases ($k = 1, 2, 3$). By grounding the reward in a relative, distributional comparison rather than raw counts, our conjugate reward model effectively denoises PV feedback, ensuring that policy optimization aligns with latent reasoning quality rather than stochastic verification artifacts. This enables stable and meaningful reinforcement learning even in the presence of imperfect process-level supervision. The complete RL training process is demonstrated in Algorithm 1.

Table 1: **Overall evaluation results for Intern-S1-MO and each baseline.** Here, the AIME2025 and HMMT2025 scores for the baseline models (first six rows) are from their respective technical reports or corresponding results in Matharena. For IMO2025, we report the pass@4 score, while the remaining benchmarks report the pass@1 score. **Bold** represents the best performance.

| Model | HMMT2025 | AIME2025 | CNMO2025 | IMO2025 |
|---|---|---|---|---|
| Gemini2.5-pro | 82.5 | 83 | 157.5 | 14 |
| o3-high | 77.5 | 88.9 | 138.5 | 12.5 |
| Grok4 | 92.5 | 91.7 | 84 | 4 |
| GPT-OSS-120B | 90 | 92.5 | 130 | 11 |
| DeepSeek-R1-0528 | 76.67 | 87.5 | 113.5 | 6.5 |
| Qwen3-235B-A22B | 60.4 | 81.5 | 109 | 14 |
| Intern-S1-mini-MO | 79.2 | 87.3 | 176.3 | 17 |
| Intern-S1-MO | **95** | **96.6** | **232.4** | **26** |

## 4 EXPERIMENT

### 4.1 EXPERIMENT SETUP

**Implementation.** We collect a set of problems from Art of Problem Solving (AoPS)[2] and in-house datasets as cold-start data, whose domain across middle school, university, and competition-level, including both solution-based and proof-based questions. We generate candidate trajectories using a variant of Intern-S1 (Bai et al., 2025), then employ the CompassVerifier Liu et al. (2025) and process verifier as the judger for solution-based and proof-based questions, respectively. Simultaneously, we chose a portion of the challenging problems as RL data, based on the pass rate of Intern-S1 on those data. Finally, built on Intern-S1 (Bai et al., 2025), we developed Intern-S1-MO, the multi-agent system solving complex reasoning problems through hierarchical decomposition. Subsequently, by distilling it, we built a lite system based on Intern-S1-Mini, called Intern-S1-mini-MO.

**Evaluation.** We use some well-established mathematical datasets for evaluation, such as AIME2025 (Mathematical Association of America), HMMT2025 Feb (Harvard–MIT Mathematics Tournament), IMO2025, and CNMO2025 (Chinese National High School Mathematics Olympiad)[3]. For CNMO2025 and IMO2025, we only evaluate the non-geometric parts. Referring to the approach of MathArena (Balunovi'c et al., 2025), we build an evaluation system that scores the answer based on fine-grained grading points (details in Appendix D), and we employ it in the evaluation for IMO2025 and CNMO2025. For each sample, we perform 16 independent rollouts and use the unbiased `pass@1` (Chen, 2021) as the metric, except for IMO2025, which we use `pass@4`.

**Baseline.** We conduct evaluations against several baselines, including Gemini2.5-pro (DeepMind, 2025a), o3-high (OpenAI, 2025a), Grok4 (xAI, 2025), GPT-OSS-120B (OpenAI, 2025b), DeepSeek-R1-0528 (Guo et al., 2025), and Qwen3-235B-A22B (Yang et al., 2025). For some benchmarks (AIME2025 and HMMT2025), we report the scores of such baseline models from their respective technical reports or corresponding results from Matharena.

### 4.2 OVERALL RESULTS

The quantitative results, summarised in Table 1, reveal a distinct performance hierarchy where our proposed framework significantly outperforms current state-of-the-art baselines. The parameter-efficient variant, Intern-S1-mini-MO, exhibits exceptional reasoning density. It surpasses all closed-source and open-weights baselines on the highly challenging CNMO2025 benchmark (scoring 176.3 compared to Gemini 2.5 Pro's 157.5) and achieves a score of 17 on IMO2025. This result suggests that the performance gains are primarily attributable to our architectural innovations, and offers compelling evidence that complex mathematical reasoning can be achieved with favorable inference-time efficiency.

---

[2] https://artofproblemsolving.com/community
[3] https://www.cms.org.cn

Table 2: **Ablation study results.** Here, "Single-round with Agents" means that only one round of inference is performed in the agent system, which is the left part of Fig 2. "+ Multi-round Reasoning" means performing a full multi-round reasoning, but without providing scores for intermediate lemmas and a revised final loop. "+ Theorem Verifier" means providing the confidence score for the intermediate lemma, that is, which is the left and middle part of Fig 2. "+ Process Verifier" means the overall inference workflow. And "+ OReal-H" means the agents are trained by the RL algorithm introduced in Section 3.

| Model | HMMT2025 | AIME2025 | CNMO2025 |
|---|---|---|---|
| Single-round with Agents | 70.8 | 81.9 | 178.0 |
| + Multi-round Reasoning | 85.4 | 91.0 | 201.7 |
| + Theorem Verifier | 86.3 | 93.3 | 203.0 |
| + Process Verifier | 89.1 | 94.0 | 215.2 |
| + OReal-H | 95.0 | 96.6 | 232.4 |

Analyzing performance deltas across benchmarks reveals a qualitative divergence in problem-solving requirements. On relatively standard competition sets like HMMT2025 and AIME2025, the gap between strong baselines and our method is present but narrower. We hypothesize that performance in these regimes is partially saturated by models capable of pattern matching and heuristic retrieval from pre-training data. On CNMO2025 and IMO2025, whose problems demand the construction of novel proof paths and the synthesis of auxiliary lemmas. Intern-S1-MO excels here precisely because it maintains a persistent logical state across rounds. Unlike single-pass models that must restart reasoning from scratch upon failure, our agent accumulates partial progress (e.g., establishing a necessary inequality or isolating a geometric invariant), effectively simulating the "scratchpad" utility used by human experts.

The performance on IMO2025 warrants specific contextualization. A score of 26 places Intern-S1-MO within the top percentile of global human competitors, outperforming the national team averages of most participating countries. Preliminary error analysis indicates that the remaining deficit largely stems from problems requiring highly idiosyncratic transformations or "spark-of-insight" constructions that elude systematic search. Collectively, these findings demonstrate that while parameter scale provides a necessary foundation, the transition from competency to mastery in Olympiad-level mathematics requires a structured, verifiable cognitive architecture capable of sustained, multi-step deduction.

### 4.3 ABLATION STUDY

To better understand the contribution of each key component in Intern-S1-MO, we conduct a systematic ablation study. Due to the limited number of problems in IMO2025 (only five), which brings the volatile results, we compare the evaluation results on HMMT2025, AIME2025, and CNMO2025.

As described in Section 2 and Section 3, the architecture of Intern-S1-MO integrates several components, including multi-round reasoning with lemma search and summary, lemma verification, process validation, and an RL framework for training the LRM using the online explored trajectories. However, it is crucial to disentangle their individual impacts to validate design choices and assess whether performance gains stem from architectural sophistication or synergistic interactions among modules. Therefore, we incrementally build up the full agent system from a simplified baseline, called "Single-round with Agents", which means that only one round of inference is performed in the agent system. Then we progressively add the corresponding component.

As shown in Table 2, we add each component step by step, where "Single-round with Agents" represents the left part of Fig 2, "+ Multi-round Reasoning" represents the left and middle part of Fig 2 without providing scores for intermediate lemmas, "+ Theorem Verifier" represents the reasoning pattern with the scored lemma, "+ Process Verifier" represents the overall inference workflow, and "+ OReal-H" represents the agent system trained by the RL algorithm.

The gradual addition of the modules steadily increases the `pass@1` scores of Intern-S1-MO on each benchmarks, proving every constituent component within the proposed framework serves a non-redundant function. Ultimately, compared to the initial baseline, our method improves the score in CNMO2025 from 178.0 to 232.4 and also achieves gains on HMMT2025 and AIME2025.

## 5 RELATED WORK

### 5.1 MATHEMATICAL REASONING AGENTS

Recent advancements in large reasoning models have significantly enhanced their performance on mathematical reasoning tasks; however, systematic exploration and reflection are still areas that require further investigation. A notable approach involves the use of tree search methods—such as Tree-of-Thoughts (Yao et al., 2023) and Monte Carlo Tree Search (Zhang et al., 2024a)—to facilitate parallel search during inference. While these methods broaden the search landscape, they often lack depth and struggle to effectively decompose complex problems(Sun et al., 2025; Balunovi'c et al., 2025). Other research has focused on augmenting LLMs with external tools to ground reasoning in computation or verified knowledge (Gou et al., 2023; Shao et al., 2024; Huang et al., 2025). Yet, these tools typically serve to enhance the existing reasoning process rather than fundamentally restructure it. More recent efforts propose structured reasoning frameworks that integrate planning, exploration, and reflection to iteratively refine solutions (Yuan & Xie, 2025). These methods outperform standard chain-of-thought prompting on challenging problems, but they usually rely on carefully designed prompts and sometimes human-provided hints. Importantly, they shift reasoning from single-path generation to structured problem solving. Yet, training math agents—where exploration and reflection are optimized through learning signals—remains an emerging area(Plaat et al., 2024). Recent initiatives have introduced structured reasoning frameworks that integrate exploration and reflection to iteratively refine solutions (Huang & Yang, 2025). These methods have been shown to outperform traditional methods on challenging problems. However, they often depend on meticulously crafted prompts and, at times, hints provided by humans.

### 5.2 REINFORCEMENT LEARNING FOR MATH AGENTS

Reinforcement learning (RL) for mathematical reasoning has primarily focused on outcome rewards, where feedback is based solely on final answer correctness. Despite this sparse signal, methods like ARTIST (Zhang et al., 2024b), ToRL (Li et al., 2025b), and rStar2-Agent (Shang et al., 2025) exhibit emergent agentic behaviors—such as adaptive tool use, self-correction, and context-aware reasoning. Scaling studies (e.g., ZeroTIR Mai et al. (2025)) further show that increased training effort leads to more sophisticated tool-integrated strategies. Nevertheless, current math agents remain limited: their decisions are mostly confined to choosing when to retry within a fixed reasoning template—rather than engaging in strategic planning or deep exploration. Critically, they lack summarization and cross-episode awareness. While approaches like TTRL (Zuo et al., 2025) and Satori (Shen et al., 2025) introduce basic reflection or meta-actions, they operate within isolated reasoning episodes and do not support cumulative knowledge transfer across inferences. Process-aware RL and verifier-guided training (e.g., Prover-Verifier Games (Kirchner et al., 2024)) aim to provide intermediate supervision with predefined rules or code execution, and are not well-suited for complex reasoning scenarios. In this paper, we use a process verifier to judge the rigor of natural language proofs, which provides a more flexible feedback signal.

## 6 CONCLUSION

This paper aims to address the critical bottleneck in large reasoning models (LRMs) for complex mathematical reasoning: the inherent limitation of context length, which has hindered progress in solving ultra-challenging tasks such as International Mathematical Olympiad (IMO) problems. To this end, this paper introduces Intern-S1-MO, an LRM-driven multi-agent system that conducts multi-round hierarchical reasoning, which conducts reasoning, summary, and verification at each round. By maintaining a compact memory in the form of lemmas, Intern-S1-MO can more freely explore the lemma-rich reasoning spaces in multiple reasoning rounds, which significantly extends the 64K constraints of LRMs by about 8 times. We further propose OREAL-H, an RL framework for training the LRM to simultaneously bootstrap the reasoning ability of the LRM and elevate the overall performance of Intern-S1-MO. Intern-S1-MO can now solve problems that require humans to think about 1.5 hours, which eventually obtains 26 out of 35 points on the non-geometry problems of IMO2025, matching the performance of silver medalists. We wish the work paves the way for future research that adopts LRMs for mathematical research.

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

## THE USE OF LARGE LANGUAGE MODELS (LLMs)

We used LLMs solely for language polishing. The scientific ideas, methodology, analyses, and conclusions were entirely developed by the authors, while the LLMs assisted only in improving clarity and readability of the text.

## A  SYSTEM PROMPTS FOR MATH AGENTS

Our workflow primarily comprises iterative policy lemma search and summarisation, alongside corresponding lemma and final answer verification. Following the final answer verification, the policy model will undergo iterative refinement based on feedback. The prompts for these five actions are presented as follows:

### A.1  LEMMA SEARCH

Listing 1: Lemma Search

```
**Objective:**
Your task is to provide a rigorous mathematical proof and solution for
    ↪ the given problem. The problem is expected to be challenging. Your
    ↪ primary goal is to demonstrate a deep and correct understanding of
    ↪ the problem through logical, step-by-step reasoning.

**Guiding Principles:**

1.  **Rigor is Paramount:**
    *   Every step in your proof must be logically sound and clearly
    ↪ justified.
    *   The final answer is secondary to the correctness of the
    ↪ derivation. A correct answer resulting from a flawed or incomplete
    ↪ proof will be considered a failure.

2.  **Embrace Partial Solutions:**
    *   It is understood that a complete solution may not be found in a
    ↪ single attempt.
    *   If you cannot provide a complete solution, you must provide any
    ↪ significant partial results that you can prove with full rigor.
    *    **Do not guess or provide solutions with logical gaps.** Instead,
    ↪  focus on what you *can* prove.
    *   Examples of valuable partial results include:
        *   Proving a key lemma.
        *   Solving one or more cases of a proof by cases.
        *   Establishing a critical property of the mathematical objects
    ↪ involved.
        *   For an optimization problem, proving an upper or lower bound.
    *   Clearly state which parts of the problem you have solved and
    ↪ which remain open. Acknowledging the limits of your solution is a
    ↪ critical part of the task.

3.  **Mathematical Formatting:**
    *   All mathematical variables, expressions, equations, and relations
    ↪  must be formatted using TeX. For example: `Let $G$ be a group and
    ↪ let $H$ be a subgroup of $G$.`

**Output Format:**
Your response MUST be structured into the following sections, in this
    ↪ exact order.

---

**1. Summary**
```

```
**a. Verdict:**
*   Begin by stating clearly whether you have found a complete or a
    ↪ partial solution.
*   **For a complete solution:** State the final answer. (e.g., "I have
    ↪ found a complete solution. The answer is...")
*   **For a partial solution:** Clearly state the main rigorous
    ↪ conclusion(s) you have proven (for example: "I have not found a
    ↪ complete solution, but I have rigorously proven the following:").
    ↪ Your output must strictly follow the Markdown and LaTeX formatting
    ↪ guidelines below:

    - **Format for Proven Lemmas:**
        - All **proven lemmas** and their proofs should be placed
    ↪ together inside a single `\boxed{}` environment.
        - Use `---` horizontal lines to separate different lemmas.
        - Each lemma should begin with `**Lemma X:**`, where `X` is a
    ↪ positive integer.
        - State each lemma concisely and formally, using LaTeX as
    ↪ appropriate.
        - The proof should immediately follow, starting with `**Proof X
    ↪ :**`.
        - Each step of the proof should use an unordered list (`*`), and
    ↪ each step should begin with `**Step Y:**`.

    - **Format for Unproven Lemmas:**
        - All **unproven lemmas** should be placed together in a separate
    ↪  `\boxed{}` environment.
        - Each lemma should begin with `**Lemma X:**`.
        - If all key steps are already provided in the "Provided Lemmas"
    ↪ section or have been fully proven (i.e., **no new unproven lemmas
    ↪ are found**), simply include `**Lemma -1**` in this box.

    - **Example Output Format:**
    ```
    \boxed{
    **lemma n+1**:{lemma n+1}
    **proof n+1**:
    *step 1:{step 1}
    *step 2:{step 2}
    *step 3:{step 3}
    ---
    **lemma n+2**:{lemma n+2}
    **proof n+2**:
    *step 1:{step 1}
    ...
    }
    \boxed{
    **withoutproof**:
    **lemma -1**
    }
    ```

    - After outputting the lemmas, you should end your response
    ↪ immediately without proceeding to the subsequent sections.

**b. Method Sketch:**
*   Provide a high-level, conceptual outline of your logical argument.
    ↪ This should be clear enough for an expert to grasp your approach
    ↪ without reading the full proof.
*   Include:
    *   A narrative of your overall strategy.
    *   The full and precise mathematical statements of any key lemmas or
    ↪  major intermediate results you proved.
    *   A description of any key constructions or case splits that form
    ↪ the backbone of your argument.
```

```
**2. Detailed Solution**

*   Present the full, step-by-step mathematical proof of your results.
*   This section should contain *only* the rigorous proof itself, free
    ↪ from any commentary, reflections on your process, or alternative
    ↪ approaches you considered.
*   The level of detail must be sufficient for an expert to verify the
    ↪ correctness of your reasoning without needing to fill in any gaps.
```

## A.2 LEMMA SUMMARIZATION

Listing 2: Lemma Summarization

```
You are a top-tier mathematical research assistant, proficient in the
    ↪ logical analysis and argumentation of high-level competitive
    ↪ mathematics.

Your core task is to conduct an in-depth analysis of a solution approach
    ↪ generated by a large language model for problems at the
    ↪ International Mathematical Olympiad (IMO) level, identifying and
    ↪ extracting all key lemmas.

During this analysis, you must rigorously distinguish between
    ↪ propositions **newly proposed** by the model and **universal lemmas
    ↪ ** already provided by us. Your final output **shall only contain**
    ↪  those lemmas appearing in the model's solution approach but not
    ↪ provided in the universal lemma repository.

**The input comprises three sections:**
1.  `### Problem ###`: The mathematical problem requiring resolution.
2.  `### Provided Lemmas ###`: A set of known, proven lemmas for
    ↪ reference during problem-solving.
3.  `### Model's Thinking Process ###`: The reasoning process generated
    ↪ by the large language model to solve the problem.

**Your output must adhere to the following principles and format:**

#### **A. Extraction Principles**

1.  **Novelty**: Extract only lemmas first introduced or proven within
    ↪ the `Model's Thinking Process`. Do not include lemmas from the `
    ↪ Provided Lemmas` if the model utilises them.
2.  **Classification**: Extract only new lemmas satisfying the following
    ↪ conditions:
    *   **Proven Lemmas**: Propositions explicitly stated or implicitly
    ↪ utilised within the `model's problem-solving approach`, accompanied
    ↪  by a complete or core proof.

#### **B. Strict Formatting Requirements**

Your output must strictly adhere to the following Markdown and LaTeX
    ↪ formatting.

1.  **Format for Proven Lemmas:**
    *   Each **proven lemma** and its proof must be placed within a
    ↪ separate, non-nested `<lemma>...</lemma>` environment, with the
    ↪ opening and closing tags each occupying a distinct line. The number
    ↪  of `<lemma>...</lemma>` environments must match the number of
    ↪ lemmas extracted in this round. Note that input lemmas may not be
    ↪ presented in this format.
    *   Each lemma must begin on a new line with the text `\n**Lemma X (
    ↪ Lemma X):**`, where `X` is a positive integer numbering. The
    ↪ opening line of this lemma must occupy a complete line. You must
```

```
      ↪ strictly adhere to this format. If the lemma has any additional
      ↪ names, annotations, or other descriptions, you may append
      ↪ explanatory text enclosed in brackets after '\n**Lemma X (Lemma X)
      ↪ :**', e.g., '\n**Lemma 2 (Lemma 2):**(Dilworth's Theorem)'.
      *   Subsequently, the remainder of this line must strictly state the
      ↪ content of the lemma. This requires a complete exposition of the
      ↪ lemma extracted from the 'Model Problem-Solving Approach'. As the '
      ↪ Model Problem-Solving Approach' frequently introduces entirely new
      ↪ symbols and notations, you must rigorously provide their
      ↪ definitions. Should these definitions involve existing lemmas, you
      ↪ must also specify which particular definitions from those existing
      ↪ lemmas are being referenced.
      *   The statement of the lemma should employ concise, formal
      ↪ mathematical language, utilising LaTeX where appropriate.
      *   This is immediately followed by the proof, commencing on a new
      ↪ line with '**Proof X:**'.
      *   Each step of the proof should begin with an unordered list '*'
      ↪ and be prefixed with '**Step Y:**'. Each step should occupy a
      ↪ separate line. Where an existing lemma is referenced in the proof,
      ↪ you must explicitly state which existing lemma is being referenced
      ↪ and how it is being applied.
      *   The positive integer numbering for each round should be the
      ↪ largest number in the general lemma library incremented by 1. Note
      ↪ that some lemmas in the general lemma library are corrected
      ↪ versions of others. These corrected lemmas share the same numbering
      ↪  as the original lemma, but are marked with the suffix '-fixed'.

Below is a sample input-output pair:

### Problem Statement (Problem) ###
{Problem}

### Provided Lemmas (Provided Lemmas) ###
<lemma>
**Lemma 1 (Lemma 1)**:
**Proof 1 (Proof 1):**:
</lemma>
---...

---
<lemma>
**Lemma n**:
**Proof n:**:
</lemma>

### Model's Thinking Process ###
{Thinking}

---

####`DESIREDOUTPUT:
<lemma>
**Lemma n+1:** lemma n+1
**Proof n+1:**:
* **Step 1:** step 1
* **Step 2:** step 2
* **Step 3:** step 3
</lemma>
<lemma>
**Lemma n+2:** lemma n+2
**Proof n+2:**:
* **Step 1:** step 1
* **Step 2:** step 2
* **Step 3:** step 3
</lemma>
```

```
...
```

## A.3 LEMMA VERIFY

Listing 3: Lemma Summarization

```
You are a mathematics and logic expert. Your task is to evaluate the
    ↪ correctness of a newly proposed lemma. This lemma relates to the
    ↪ main mathematical problem and may rely on a provided library of
    ↪ existing lemmas.

Your goal is to meticulously check the proof of the new lemma, step by
    ↪ step, to identify the index of the first incorrect step. The index
    ↪ starts at 0 for the first step. If the proof is entirely correct,
    ↪ you should output -1.

Instructions:

- You will be given:
  1. The Main Question: The overarching problem providing context.
  2. Provided Lemmas: A library of existing statements assumed to be
     ↪ correct.
  3. The New Lemma and Its Proof: The student's work to be evaluated,
     ↪ with the proof skeleton broken down into steps.

- A key part of your evaluation is to verify that any use of a lemma from
    ↪  the Provided Lemmas library is correctly applied and that its
    ↪ preconditions are satisfied. The logical inferences within the
    ↪ proof must be sound and either self-evident, derived from the main
    ↪ question's conditions, or justified by one of the provided lemmas.

- You must perform a step-by-step check of the entire solution. Present
    ↪ this analysis as a Detailed Verification Log:
  - Use a numbered list; each item corresponds to a step in the student's
    ↪  proof.
  - For correct steps, provide a brief justification.
  - For steps with errors or gaps, provide a detailed explanation.
  - Do not use a table.

- Finally, at the conclusion of your response, always include a First
    ↪ Error, formatted as '\box{{STEPk}}', where 'k' denotes the index of
    ↪  the first incorrect step. For instance, if step 2 is incorrect,
    ↪ respond with '\box{{STEP2}}'. Should all steps be correct, respond
    ↪ with '\box{{STEP-1}}'.

- The new lemma to verify is guaranteed to possess both a proposition and
    ↪  a proof skeleton. Should any component be missing (thus rendering
    ↪ it an invalid lemma), directly output 'FORMAT_ERROR' followed by a
    ↪ description of the observed error. In such instances, no further
    ↪ output is required; omit the '\box{{STEP}}' indicator.

---
### Question ###:
{Question}

### Historical Lemma Repository (Provided Lemmas) ###
{ProvidedLemmas}

### New Proof to Verify ###
{NewLemmatoVerify}

### Detailed Verification Log and First Error ###:
```

## A.4 FINAL ANSWER VERIFY

Listing 4: Final Answer Verify

```
You are a mathematics and educational expert tasked with evaluating the
    ↪ correctness of a student's answer. The student's solution is broken
    ↪ down into steps, and your goal is to identify the index of the
    ↪ first incorrect step. The index starts at zero for the first step.
    ↪ If all steps are correct, you should output -1.

Instructions:
- You will receive a question along with the student's answer, divided
    ↪ into steps. Each step is presented in a separate paragraph.
- You are encouraged to express your internal reasoning within <think
    ↪ >...</think> tags. After presenting your thinking process, you **
    ↪ must** write a summary of your evaluation. Finally, at the end of
    ↪ your response, always include an integer within \\box{{STEP}}. For
    ↪ example, if step 2 is incorrect, respond with \\box{{STEP2}}. If
    ↪ all steps are correct, respond with \\box{{STEP-1}}.
- The student's answer may involve a number of indexed lemmas with their
    ↪ proofs. If you found any of them incorrect, you should report that.
    ↪ For example, if lemma 3 is incorrect, respond with \\box{{LEMMA3}}
    ↪ instead of \\box{{STEP3}}. If all lemmas are correct, you should
    ↪ them detect any incorrect steps. If everything is fine, respond
    ↪ with \\box{{STEP-1}}. Do not count steps inside a lemma. The step
    ↪ index depends on the detailed solution part.
- Some steps may initially appear incorrect but are later corrected in
    ↪ subsequent steps. If a reflection or revision is both accurate and
    ↪ reasonable, the step should be considered correct. If there are
    ↪ multiple reflections, consider only the final one.
- In cases where the problem is ambiguous, consider all possible
    ↪ interpretations and determine if the student's response aligns with
    ↪ any of them.
- Evaluate the entire solution, as some intermediate steps might seem
    ↪ incorrect initially but are rectified later, such as dismissing an
    ↪ extraneous root. Ensure you consider the entire context and, if
    ↪ necessary, review the steps more than once.
- The errors to identify can be very subtle, sometimes hiding in the
    ↪ inexplicit applications of theorems or conditions. So you should
    ↪ actively checking every small logical inferences at a small
    ↪ granularity carefully, either in natural language or in formulas.
- If an error does not affect the overall reasoning, or an gap can be
    ↪ recovered by your effort, you should not report them as incorrect.
- To help you identify the possible errors, every first time you checking
    ↪ a step, you should repeat it in case you missed subtle information
    ↪ . Then you should check its validity by examing its logical
    ↪ inferences within the step/sentences/subsentences one by one.
- Every step should have solid logical basis. Guessing without proof is
    ↪ not allowed.

---
**Question**: {question}
**Reference Answer**: {reference}
**Student's Answer**: {response}
**First Error**:
```

## A.5 SELF-IMPROVE WITH VERIFY FEEDBACK

Listing 5: Self-improve with Verify Feedback

```
You are a mathematics and logic expert. Your task is to improve a
    ↪ solution to a math Olympaid problem given a presented solution
    ↪ trial some comments about the solution.

Your goal is to get a concised, improved solution that solve all errors
    ↪ reasonably pointed out by comments, fill all gaps mentioned by
    ↪ comments and defend other issues that is defendable. Besides, you
    ↪ should compress the complexity of the solution and try your best to
    ↪  make it highschool-level.

Instructions:

- You will be given:
  1. The Main Question: The overall problem providing context.
  2. Provided Solution: The student's work to be evaluated, with a
     ↪ verdict and the complete solution divided into steps.
  3. Previous Comments: Prior attempts to detect specific types of errors.
     ↪  Treat these as helpful guidance rather than authoritative. If they
     ↪  flag something, fix or defend.

- You must present a improved solution with SAME FORMAT. Typically a
    ↪ solution comes with a summary section and a detailed solution
    ↪ section.

- You are free to decide the idea/approach of your improved solution. You
    ↪  can just fix specific issues, restructure certain parts of the
    ↪ previous answer, or even discard the original solution if
    ↪ considered as unfixable.

- You are adviced to use highschool level of math. If you choose to use
    ↪ university level, then you should treat the readers as smart
    ↪ hihgschool students with no backgrounds, then provide the specific
    ↪ introduction of certain knowledge needed.

- If the solution to improve contains NO USEFUL INFORMATION (e.g. simply
    ↪ admitting its failure). Then you should just return "I have not
    ↪ found a complete solution".
---
### Question
{Question}

### Solution to Improve
{SolutiontoVerify}

### Previous Comments
{PreviousCheckingEfforts}

### Your Improved Solution
```

## B  IMPLEMENTATION DETAILS

### B.1  INFERENCE BUDGET

Our agentic system is a scalable framework that allows for custom inference budgets based on problem difficulty. Theoretically, a higher inference budget leads to better performance, which aligns with the core logic of the TTS (test time scaling) strategy (Muennighoff et al., 2025).

To control evaluation costs, we set some default inference budgets. Specifically, we set the maximum number of inference rounds for the reasoner and summarizer agent to 8, the number of parallel verifications for the theorem verifier to 4 for each lemma, and the maximum number of rounds for final iterative revision based on the process verifier to 8. For the reasoner and summarizer agent, the max length of output is set to 64k.

---

**Algorithm 1** Evolution of Hierarchical Math Agents via Lemma Dependency Graphs

---

**Require:** Policy $\pi_\theta$, Verifier (PV) $V_\phi$, Dataset $\mathcal{D}$, Hyperparameters $\gamma, \alpha$
**Require:** Conjugate Reward function $R_{\text{conj}}(k, n)$ (Eq. 7)
1: **while** not converged **do**
2:     Initialize batch buffer $\mathcal{B} \leftarrow \emptyset$
3:     **for** each problem $q \in \text{Batch}(\mathcal{D})$ **do**
4:         **// Step 1: Multi-round Rollout**
5:         Sample $K$ trajectories $\{\tau^{(i)}\}_{i=1}^K$ using $\pi_\theta(\cdot|q)$
6:         **// Step 2: Process Verification**
7:         Evaluate each trajectory $\tau^{(i)}$ via $V_\phi$
8:           Compute final reward $R^{(i)} \leftarrow R_{\text{conj}}(k, n)$
9:         **// Step 3: Construct Lemma Dependency Graph**
10:        Initialize graph $\mathcal{G} = (\mathcal{V}, \mathcal{E})$ with lemmas from all $\{\tau^{(i)}\}$
11:       Identify terminal nodes where $R^{(i)} \neq 0$
12:       Backpropagate values recursively to compute lemma values:
13:         $v(l) \leftarrow \mathbb{E}_{l' \in \text{Succ}(l)}[v(l')]$                   ▷ Eq. (4)
14:       **// Step 4: Compute Progress-Conditioned Advantage**
15:       **for** each trajectory $\tau^{(i)}$ and step $t = 1 \ldots T$ **do**
16:         **if** $C_t^{(i)} = 1$ **then**                 ▷ Valid progress round
17:           Estimate state value: $V(s_t) \leftarrow \max_{l \in \mathcal{L}_t} v(l)$
18:           Compute next-state value: $V(s_{t+1}) \leftarrow \max_{l' \in \mathcal{L}_{t+1}} v(l')$
19:           Calculate advantage $A_t^{(i)}$ using TD error:
20:            $A_t^{(i)} \leftarrow r_t + \gamma V(s_{t+1}) - V(s_t)$          ▷ Eq. (5)
21:         **else**
22:           $A_t^{(i)} \leftarrow 0$             ▷ Mask non-progress rounds
23:         **end if**
24:         Add $(s_t, u_t, A_t^{(i)})$ to $\mathcal{B}$
25:       **end for**
26:     **end for**
27:     Optimization via OReal Loss (Lyu et al., 2025)
28: **end while**

---

## B.2   HYPERPARAMETERS DETAILS

During training iterations, each batch consists of 64 questions, with 16 rollouts per question. The max length of each rollout trajectory is set to 65536 tokens. Then the correctness of each response is averaged to calculate the pass rate, and questions with an overall pass rate of 0 or 1 are discarded.

For optimization, the policy model is trained with a learning rate of $5e-7$. Both models employ a cosine annealing learning rate schedule, decaying to $1/5$ of the initial learning rate over time. We optimize both models using the AdamW optimizer. The KL coefficient $\beta$ is set to 0.01.

## C   OREAL-H ALGORITHM

The complete RL training procedure is described in Algorithm 1.

## D  GRADING DETAILS

Automated evaluation of complex mathematical proofs presents substantial challenges. LLMs often exhibit excessive sensitivity to lexical phrasing while occasionally overlooking missing logical reasoning. To bridge the gap between automated evaluation pipelines and human experts, we designed a fine-grained grading scheme tailored to the nature of the problem.

**Calculation-Centric Evaluation (HMMT, AIME)**    For datasets primarily focused on final answers, such as HMMT and AIME, we only employ **Final Answer Accuracy** as the sole metric. A response is awarded full score if and only if the extracted final answer matches the ground truth exactly; otherwise, it receives zero.

**Proof-Oriented Evaluation (CNMO, IMO)**    For Olympiad-level proof problems (e.g., CNMO and IMO), we adopt a rubric-based scoring logic inspired by MathArena Balunovi'c et al. (2025), with critical modifications to ensure rigor.

The key difference here is their grading schemes often list only the necessary sub-propositions, lacking explicit constraints on the derivation of conclusions. When used with LLM-based judges, this ambiguity frequently leads to significant false positives. To rectify this, we augmented the grading scheme by explicitly coupling sub-propositions with their corresponding conclusion requirements. A representative example of our revised grading scheme is shown in FIgure. 3.

To mitigate the inherent stochasticity of LLM judges, we implement an ensemble evaluation protocol. Each generated solution is evaluated in parallel across $N = 8$ independent runs. For grading points worth more than 1 point, the model is awarded partial credit if it provides a valid partial proof. The final score for a solution is calculated as the arithmetic mean of the total scores obtained across the eight evaluation runs.

```
[
  {
    "desc": "1 point should be given for rigorously describing a
    ↪ construction for $n$=3. Should prove that $k=2$ is impossible, and
    ↪ that $k=0,1,3$ is possible.",
    "points": 1,
    "title": "Describing a construction for $k=0,1,3$ for $n=3$"
  },
  {
    "desc": "1 point should be given for just finding the answer $k=0,1,3
    ↪ $ is valid for all $n$. Providing a specific construction is
    ↪ sufficient to earn points.",
    "points": 1,
    "title": "Reaching the answer $k=0,1,3$ for all $n$"
  },
  {
    "desc": "Stating and proving that the perimeter sides ($x=1, y=1, x+y
    ↪ =n+1$) contain a total of $3n-3$ points.",
    "points": 1,
    "title": "Making an statement about the boundary points"
  },
  {
    "desc": "Arguing that any sunny line can cover at most two points on
    ↪ the perimeter sides, so for $n>3$, there must be at least one non-
    ↪ sunny line covering a complete boundary line.",
    "points": 2,
    "title": "Proving the existence of a non-sunny line covering a
    ↪ complete boundary line"
  },
  {
    "desc": "Stating and proving that if a non-sunny line contains one of
    ↪  the 3 perimeter sides ($x=1, y=1, x+y=n+1$), the problem can be to
    ↪  reduce for $n-1$ without changing the answer.",
    "points": 1,
    "title": "Reducing the problem from $n$ to $n-1$ given a boundary
    ↪ line"
  },
  {
    "desc": "Finishing by summarizing the final answer that for any $n$,
    ↪ the possible values of $k$ are 0, 1, and 3.",
    "points": 1,
    "title": "Finishing"
  }
]
```

Figure 3: **Example of the Refined Grading Scheme for IMO2025 P1.** This JSON structure outlines the specific proof obligations, point allocation, and partial credit policies used to guide the LLM judge.

# E  LEMMA GRAPH

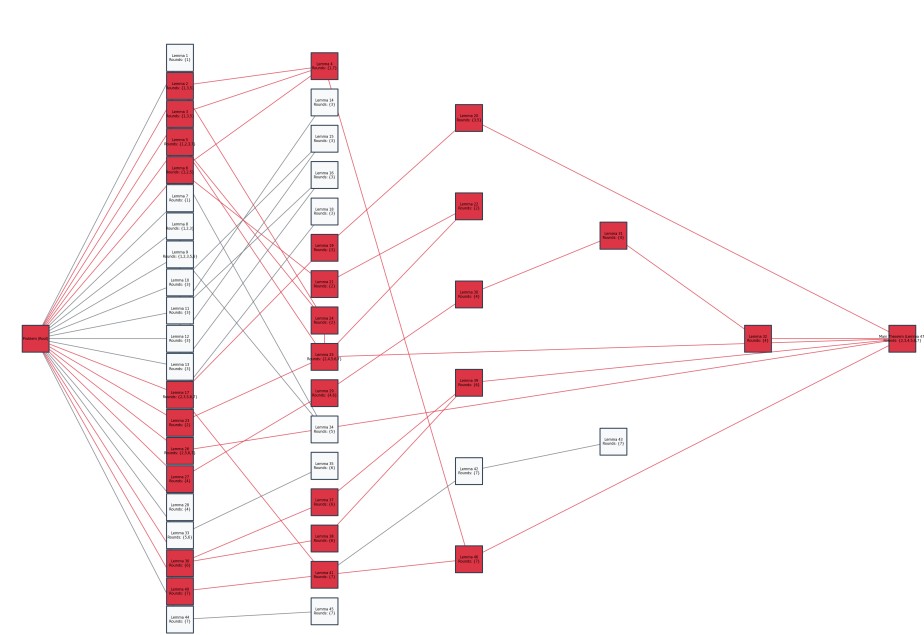

Figure 4: **Example of a lemma graph.** Red nodes mark all lemmas contributing to the final conclusion, with the numbers below indicating the number of rollout in which each lemma appeared.

## F CASE EXAMPLE

> **Case Study: AMO-Bench P1**
>
> **Problem Statement:** Divide each side of an equilateral triangle $ABC$ into four equal parts. Through each division point, draw lines parallel to the other two sides, defining 15 lattice points. Find the smallest integer $n$ such that, no matter which $n$ points are chosen, there always exist three points that form an isosceles triangle.

▷ **Reasoning Process & Lemma Accumulation**

**Round 1**

**Lemma 1:**
The 15 lattice points obtained by dividing each side of an equilateral triangle $ABC$ into four equal parts can be uniquely represented by integer pairs $(i, j)$, where

$$i \geq 0, \qquad j \geq 0, \qquad i + j \leq 4.$$

Let $P = (i_1, j_1)$ and $Q = (i_2, j_2)$ be any two points. Their squared Euclidean distance satisfies

$$|PQ|^2 = \Delta i^2 + \Delta i \, \Delta j + \Delta j^2, \qquad \text{where } \Delta i = i_1 - i_2, \ \Delta j = j_1 - j_2.$$

**Proof 1:**

- **Step 1:** Establish a coordinate system with vertex $A$ as the origin, and basis vectors $\mathbf{u} = (1, 0)$ and $\mathbf{v} = \left(\frac{1}{2}, \frac{\sqrt{3}}{2}\right)$.

- **Step 2:** The Cartesian coordinates of any point $(i, j)$ are $i\mathbf{u} + j\mathbf{v} = (i + \frac{j}{2}, \frac{\sqrt{3}}{2}j)$.

- **Step 3:** For two points $(i_1, j_1)$ and $(i_2, j_2)$, calculate the difference vector and apply the Euclidean distance formula. Simplification yields

$$|PQ|^2 = (\Delta i + \tfrac{\Delta j}{2})^2 + \left(\tfrac{\sqrt{3}}{2}\Delta j\right)^2 = \Delta i^2 + \Delta i \, \Delta j + \Delta j^2.$$

(The confidence score is 0.75.)

**Lemma 2:**
Let $P = (a, 0)$ and $Q = (b, 0)$ be two lattice points on the same side $AB$ (where $0 \leq a < b \leq 4$). Then, any lattice point $X = (i, j)$ (satisfying $i, j \geq 0$, $i + j \leq 4$) satisfies

$$|XP| = |XQ| \quad \Longleftrightarrow \quad 2i + j = a + b.$$

**Proof 2:**

- **Step 1:** Based on Lemma 1, write $|XP|^2 = (i - a)^2 + (i - a)j + j^2$ and $|XQ|^2 = (i - b)^2 + (i - b)j + j^2$.

- **Step 2:** Subtract the two equations and cancel common terms to obtain $(i - a)^2 - (i - b)^2 + (i - a)j - (i - b)j = 0$.

- **Step 3:** Simplify to $(b - a)(2i + j - a - b) = 0$. Since $a \neq b$, we obtain the required linear relation $2i + j = a + b$.

(The confidence score is 0.0.)

......

**Lemma 10:**
Among all 15 lattice points, the cardinality of the largest point set containing no isosceles triangles is 5, and only sets of the form "all 5 lattice points of one side" can achieve this upper bound.

**Proof 10:**

- **Step 1:** By Lemma 8, taking all 5 lattice points of any single side yields an isosceles-free set of size 5.

- **Step 2:** If we attempt to construct an isosceles-free set of size 6, there are two cases:

  - It contains internal lattice points: This contradicts Lemma 6.
  - It contains only boundary lattice points: This contradicts Lemma 9.

  Thus, no isosceles-free set of size 6 exists.

- **Step 3:** Therefore, the maximum cardinality is 5, and the only maximal construction is the set of all lattice points on a single side.

(The confidence score is 0.0.)

---

## Round 2

Can not summary any new lemmas.

---

## Round 3

**Lemma 11:**
In the set of triangular lattice points, for any two points $P$ and $Q$ chosen from two different sides, there exists a unique third point $R$ (located on the third side or in the interior) such that $|RP| = |RQ|$.

**Proof 11:**

- **Step 1:** Let $P$ and $Q$ be located on sides $AB$ and $AC$, respectively. Denote their coordinates as $P = (a, 0)$ and $Q = (0, b)$, where $0 \leq a, b \leq 4$.

- **Step 2:** According to Lemma 1, calculate the difference in squared distance between the two points:

$$|RX|^2 - |RY|^2 = (i - a)^2 + (i - a)j + j^2 - (i)^2 - (i)(j - b) - (j - b)^2,$$

and simplify to obtain the equidistance condition:

$$2i + j = a + b.$$

- **Step 3:** Under the constraints $i, j \geq 0$, $i + j \leq 4$, the equation $2i + j = a + b$ has a unique integer solution $(i_0, j_0)$ within the triangle. If this solution satisfies $j_0 > 0$, then the point $R = (i_0, j_0)$ is located on side $AC$; if $j_0 = 0$, then $R$ is located on side $BC$. In any case, $R$ and $P, Q$ form an isosceles triangle.

- **Step 4:** It can similarly be proven that if $P, Q$ are located on any two different sides, a unique point $R$ satisfying the equidistance condition can be found.

- **Step 5:** In summary, for any two points on different sides, there must exist a unique third point equidistant to them, completing the proof.

(The confidence score is 0.0.)

**Lemma 12:**

In any set consisting only of points on the three sides, if the cardinality of the set is 6, then the distribution of points across the sides must be such that no more than two points lie on any single side; otherwise, an isosceles triangle must exist.

**Proof 12:**

- **Step 1:** Let set $S$ contain $x$ points from the sides, where $x \geq 3$. If any side contains at least three points, select any two points $P, Q$ on that side. By Lemma 4, there exists a third point $R$ (not on that side) such that $|RP| = |RQ|$, thus forming an isosceles triangle.

- **Step 2:** Therefore, to avoid an isosceles triangle, the number of points on any single side must be at most 2.

- **Step 3:** Assume the cardinality of $S$ is 6. The distribution across the three sides must be $(2, 2, 2)$ or $(2, 3, 1)$ (partitioned by the three sides). If $(2, 3, 1)$ occurs, the side containing three points must generate an isosceles triangle, a contradiction. Thus, the only possible distribution is $(2, 2, 2)$.

- **Step 4:** For the $(2, 2, 2)$ configuration, let two points be located on different sides, denoted $(a, 0)$ and $(0, b)$. By Lemma 11, a unique third point $R$ satisfying $|RP| = |RQ|$ can be determined. This point must lie on the third side, and thus must be in the set $S$, resulting in an isosceles triangle.

- **Step 5:** Therefore, any set of size 6 consisting only of boundary points must contain an isosceles triangle, completing the proof.

(The confidence score is 0.0.)

## Round 4

**Lemma 13:**

In any set of 6 points consisting only of points on the three sides (i.e., containing no internal points), the arrangement wherein at most two points are on any single side cannot avoid an isosceles triangle. In other words, if a point set $S \subseteq \{(i, j) \mid i \geq 0, \ j \geq 0, \ i + j \leq 4\}$ contains 6 points and no internal points, then there must exist three points $P, Q, R \in S$ such that $|RP| = |RQ|$, thereby forming an isosceles triangle.

**Proof 13:**

- **Step 1:** Let $x, y, z$ be the number of points in $S$ on sides $AB, AC, BC$, respectively, such that $x + y + z = 6$. If the number of points on any side is $\geq 3$, select any two points $P, Q$ on that side. By Lemma 4 (two points on the same side must have a third equidistant point), there exists a point $R \neq P, Q$ (located on another side or internally) such that $|RP| = |RQ|$. If $R \in S$, an isosceles triangle is formed. If $R \notin S$, then $S$ contains at most all 5 points of that side, and since $x \geq 3$ and $x \leq 5$, the possible 6-point distributions must be unbalanced, such as $(3, 2, 1), (3, 1, 2), (4, 1, 1), (5, 1, 0)$, etc. Upon checking these distributions, the equidistant point $R$ corresponding to $P, Q$ can always be found on another side, and since that side has at least one point selected into $S$, $R$ must be in $S$, resulting in an isosceles triangle.

- **Step 2:** Therefore, to avoid an isosceles triangle, the number of points on each side must be $\leq 2$. Since $x + y + z = 6$, the only possible distribution is $(2, 2, 2)$.

- **Step 3:** Assume the two points on side $AB$ are $P = (a, 0), Q = (b, 0)$ $(a < b)$. The equidistant point satisfies the line equation $2i + j = a + b$ (Lemma 2). Consulting the table (from Lemma 4), this equation has at least one solution $R$ within the triangle, and $R$ must be on another side (if $a + b$ is odd, $R$ is not on $AB$). Similarly, the point pair on side $AC$ yields the equation $i + 2j = c + d$, and the pair on side $BC$ yields $i - j = e + f - 4$ (a corollary of Lemma 4).

- **Step 4:** Since there are exactly two points on each side, there must exist a pair of points such that their equidistant point $R$ falls onto a selected point on another side. Specifically, if $a + b$ is odd, the unique integer solution to $2i + j = a + b$ lies on $BC$, $R = (\frac{a+b-1}{2}, \frac{a+b+1}{2})$, and since $BC$ contains exactly two points, one of them must be $R$. If $a + b$ is even, the midpoint $(\frac{a+b}{2}, 0)$ is on $AB$, but since only two points were selected on $AB$, $\frac{a+b}{2}$ must not equal $a$

or $b$. The equidistant point must then lie on $BC$ or $AC$, and similarly, it must fall into the already selected point set. Consequently, an isosceles triangle must appear.

- **Step 5:** In summary, any set of 6 points containing no internal points must include an isosceles triangle.

(The confidence score is 0.0.)

---

**Final Answer:**

**The Triangular Lattice**

We place the equilateral triangle $ABC$ of side length 4 such that its vertices are:

$$A = (0,0), \qquad B = (4,0), \qquad C = \left(2, 2\sqrt{3}\right).$$

Let the basis vectors for the lattice be:

$$\mathbf{u} = (1,0), \qquad \mathbf{v} = \left(\tfrac{1}{2}, \tfrac{\sqrt{3}}{2}\right).$$

Every point obtained by the construction is a lattice point, representable by integer coordinates $(i,j)$:

$$(i,j) = i\mathbf{u} + j\mathbf{v} = \left(i + \tfrac{j}{2}, \tfrac{\sqrt{3}}{2}j\right), \qquad i,j \in \mathbb{Z}_{\geq 0}, \ i+j \leq 4. \tag{1}$$

Thus, the 15 points constitute the set:

$$\mathcal{L} = \{(i,j) \mid i,j \geq 0, \ i+j \leq 4\}.$$

For two points $P = (i_1, j_1)$ and $Q = (i_2, j_2)$, the squared Euclidean distance is obtained by substituting coordinates from (1) and simplifying:

$$|PQ|^2 = (i_1 - i_2)^2 + (i_1 - i_2)(j_1 - j_2) + (j_1 - j_2)^2. \tag{2}$$

**Interior Points Force an Isosceles Triangle**

The interior points of $\mathcal{L}$ (those not on the boundary) are:

$$I_1 = (1,1), \qquad I_2 = (1,2), \qquad I_3 = (2,1).$$

Using the distance formula (2), one verifies that any interior point possesses **exactly four distinct squared distances** to the remaining fourteen lattice points:

| Point | Possible $|P - Q|^2$ |
|---|---|
| $(1,1)$ | 1, 3, 4, 7 |
| $(1,2)$ | 1, 3, 4, 7 |
| $(2,1)$ | 1, 3, 4, 7 |

Since an interior point can be at at most four different distances from all other points of $\mathcal{L}$, consider a set $S \subseteq \mathcal{L}$ that contains an interior point $I$, with $|S| = 6$. The five other points of $S$ yield five distances from $I$. By the **Pigeonhole Principle**, two of these distances must be equal. These two points, together with $I$, form an isosceles triangle. Consequently, **any six-point set that contains an interior point necessarily contains an isosceles triangle**.

**Six Boundary Points Also Contain an Isosceles Triangle**

We now restrict our attention to the case where all six chosen points lie on the boundary of the triangle. The boundary points lie on the three sides:

$$\begin{aligned} AB &: \ (i,0), \ i = 0,1,2,3,4, \\ AC &: \ (0,j), \ j = 0,1,2,3,4, \\ BC &: \ (i, 4-i), \ i = 0,1,2,3,4, \end{aligned} \tag{3}$$

**A Useful Fact: Equidistant Third Point**

Let $P = (a,0)$ and $Q = (b,0)$ be two distinct points on side $AB$ with $a < b$. A point $R = (i,j)$ is equidistant from $P$ and $Q$ if and only if:

$$2i + j = a + b. \tag{4}$$

Within the constraints of the lattice (i.e., $i,j \geq 0$, $i+j \leq 4$), the unique integer solution for $R$ is:

$$R = \begin{cases} (0, a+b), & a+b \leq 4 \quad \text{(lies on } AC), \\ (a+b-4, \ 8-(a+b)), & a+b \geq 4 \quad \text{(lies on } BC). \end{cases} \tag{5}$$

This demonstrates that **any two points on a side determine a single lattice point on a different side that is equidistant from them**. This property holds symmetrically for the other two sides.

**Distribution Case Analysis**

Let $S$ be a set of six boundary points. We define the counts of points on each side as:

$$x = |S \cap AB|, \qquad y = |S \cap AC|, \qquad z = |S \cap BC|.$$

Since the three vertices are counted in two sets each, we have the inequality:

$$x + y + z = |S| + (\text{number of chosen vertices}) \geq 6. \tag{6}$$

As each side contains only five points, $x, y, z \leq 5$. Up to permutation, the only possibilities for the distribution $(x, y, z)$ that satisfy $x + y + z \geq 6$ and $x, y, z \leq 5$ are:

$$\begin{aligned} &(5, 1, 0), \ (4, 2, 0), \ (4, 1, 1), \\ &(3, 3, 0), \ (3, 2, 1), \ (2, 2, 2). \end{aligned} \tag{7}$$

We prove that in every case, an isosceles triangle is unavoidable by exhibiting a pair of points on the same side whose equidistant third point (given by (5)) also belongs to $S$.

- **Case** $(5, 1, 0)$ **(or permutation):** All five points of side $AB$ are present; the sixth point lies on $AC$, say $(0, k)$ with $1 \leq k \leq 4$. Take the pair $(0, 0)$ and $(k, 0)$ on $AB$. Their sum is $k \leq 4$, so by (5) the third equidistant point is $(0, k)$, which is exactly the chosen point on $AC$. An isosceles triangle appears.

- **Case** $(4, 2, 0)$ **(or permutation):** Four points on $AB$ and two on $AC$. Let the two points on $AC$ be $(0, c)$ and $(0, d)$ with $c < d$. The pair $(0, 0)$ and $(d, 0)$ on $AB$ has sum $d \leq 4$; the third equidistant point is $(0, d)$, which must belong to $S$. An isosceles triangle is formed.

- **Case** $(4, 1, 1)$ **(or permutation):** Four points on $AB$, one on $AC$, and one on $BC$. If the point on $AC$ is $(0, k)$ with $k \leq 4$, the pair $(0, 0)$ and $(k, 0)$ on $AB$ yields the third point $(0, k)$ on $AC$. Since $(0, k) \in S$, an isosceles triangle occurs.

- **Case** $(3, 3, 0)$ **(or permutation):** Three points on $AB$ and three on $AC$. Let the points on $AB$ be $(a, 0), (b, 0), (c, 0)$ and the points on $AC$ be $(0, p), (0, q), (0, r)$. The three sums $a + b, a + c, b + c$ are distinct numbers in $\{1, \ldots, 7\}$. If any of these sums equals $p, q$, or $r$, we are done by (5). The three numbers $p, q, r$ are also distinct. Since there are three possible pairs on $AB$ and three chosen points on $AC$, by the **Pigeonhole Principle** applied to the set of sums/targets, one of the pairs on $AB$ must have its equidistant point coinciding with one of the chosen points on $AC$. Hence, an isosceles triangle exists.

- **Case** $(3, 2, 1)$ **(or permutation):** Three points on $AB$, two on $AC$, and one on $BC$. If we consider pairs on $AC$ or $BC$, similar logic applies. The most direct approach is to consider the two points on $AC$ and the single point on $BC$. The unique equidistant point must fall onto one of the three points on $AB$, proving that an isosceles triangle is unavoidable.

- **Case** $(2, 2, 2)$**:** Two points on each side. Let the points on $AB$ be $(a, 0), (b, 0)$ with $a < b$. If $a + b \leq 4$, the third equidistant point is $(0, a + b)$ on $AC$. Since $AC$ already contains two points, one of them must be $(0, a + b)$. If $a + b \geq 5$, the third equidistant point is $(a + b - 4, 8 - (a + b))$ on $BC$. Since $BC$ contains two points, one of them must coincide with this third point. Thus, an isosceles triangle is unavoidable in the $(2, 2, 2)$ case.

Since all six possibilities lead to an isosceles triangle, **any set of six boundary points contains an isosceles triangle**.

**Sharpness and Conclusion**

The five points on a single side, for example:

$$\{(0, 0), (1, 0), (2, 0), (3, 0), (4, 0)\},$$

are collinear. They contain no non-degenerate triangle at all, and therefore no isosceles triangle. Hence, a set of five points can avoid the required configuration.

Every set of six of the fifteen lattice points necessarily contains three points forming an isosceles triangle, while a set of five points may avoid it. Therefore, the smallest integer $n$ with the required property is:

$$\boxed{6}.$$

**A Failure Case before RL Training**

**Distance Formula**

Let $P_{j_1,k_1}$ and $P_{j_2,k_2}$ be two points. We set $\Delta j = j_2 - j_1$ and $\Delta k = k_2 - k_1$. From the coordinate definition, $P_{j,k} = \left( \frac{2j+k}{8}, \frac{\sqrt{3}\,k}{8} \right)$, we have:

$$\Delta x = \frac{2\Delta j + \Delta k}{8}, \qquad \Delta y = \frac{\sqrt{3}\,\Delta k}{8}.$$

Hence, the squared distance is:

$$|P_{j_1,k_1} P_{j_2,k_2}|^2 = \Delta x^2 + \Delta y^2 = \frac{(2\Delta j + \Delta k)^2 + 3\Delta k^2}{64}$$

$$= \frac{4\big(\Delta j^2 + \Delta j \Delta k + \Delta k^2\big)}{64} = \frac{\Delta j^2 + \Delta j \Delta k + \Delta k^2}{16}. \tag{4}$$

Thus, the scaled squared distances are the integers:

$$\Delta j^2 + \Delta j \Delta k + \Delta k^2, \qquad (\Delta j, \Delta k) \neq (0,0),\ |\Delta j| + |\Delta k| \leq 4. \tag{5}$$

A short enumeration of all admissible $(\Delta j, \Delta k)$ pairs gives the set of possible scaled squared distances:

$$\{\,\Delta j^2 + \Delta j \Delta k + \Delta k^2\,\} = \{1, 3, 4, 7, 9, 12, 13, 16\}. \tag{6}$$

Consequently, there are exactly eight distinct distances (up to the scaling factor $\frac{1}{16}$) between points of $\mathcal{L}$.

**Isosceles-Free Sets**

Three points form an isosceles triangle if and only if one of them is equidistant from the other two. Therefore, a set $\mathcal{S} \subseteq \mathcal{L}$ contains **no** isosceles triangle precisely when, for every $P \in \mathcal{S}$, the distances $\{|PQ| : Q \in \mathcal{S} \setminus \{P\}\}$ are all different.

Since (6) lists only eight possible distances, a point can be the apex of at most eight distinct isosceles triangles. Hence, by the Pigeonhole Principle, a set of ten points must contain an isosceles triangle. This yields the preliminary bound $n \leq 10$.

**An Isosceles-Free Set of Size Eight**

Consider the set $S$ defined as the union of all points on side $AB$ and three non-adjacent points on side $AC$:

$$S = \underbrace{\{(j,0) \mid j = 0, 1, 2, 3, 4\}}_{\text{the side } AB} \cup \underbrace{\{(0,k) \mid k = 2, 3, 4\}}_{\text{three points on } AC \text{ not including } (0,0) \text{ and } (0,1)}. \tag{8}$$

$S$ contains $5 + 3 = 8$ points. We verify that $S$ is isosceles-free.

- *Case 1: Apex $P = (i,0) \in S$ (a point on $AB$).* The distances from $P$ to the three points of $AC$ are proportional to $i^2 - ik + k^2$ for $k \in \{2, 3, 4\}$. For each fixed $i \in \{0, \ldots, 4\}$, a direct check confirms these three numbers are distinct. Distances to other points of $AB$ are $\frac{(i-i')^2}{16}$ with $i' \neq i$, which are also distinct. All distances from $P$ to the other points of $S$ are thus different.

- *Case 2: Apex $P = (0,k) \in S$ (a point on $AC$).* The same calculation, with the roles of $i$ and $k$ interchanged, shows that the distances from $P$ to the points of $AB$ are pairwise distinct. Distances to the other two points of $AC$ are $\frac{(k-k')^2}{16}$ with $k' \neq k$, which are also distinct. All distances from $P$ are different.

Thus, no point of $S$ is equidistant from two others, proving that $S$ contains **no** isosceles triangle. Consequently, the maximal size of an isosceles-free subset is $\geq 8$, implying $n \geq 9$.

**No Isosceles-Free Set of Size Nine**

Assume, for contradiction, that $\mathcal{T} \subseteq \mathcal{L}$ is isosceles-free and $|\mathcal{T}| = 9$.

**Contradiction via Internal Points and Boundaries**

- **Boundary Intersection Constraint:** If $\mathcal{T}$ contains points on all three sides ($AB$, $AC$, and $BC$), at least one side must contain three points. The distances from vertex $A$ to the five points of side $BC$ are proportional to $\{1, 1, 12, 13, 13\}$. Hence, any three points of $BC$ contain two with the same distance from $A$; together with $A$, they form an isosceles triangle, contradicting the hypothesis. Therefore, $\mathcal{T}$ is contained in the union of at most two sides.

- **Reduction:** By symmetry, we may assume $\mathcal{T} \subseteq AB \cup AC$.

- **Exclusion of $(0,1)$:** If $(0,1) \in \mathcal{T}$, then $|(0,1)(0,0)|^2 = |(0,1)(1,0)|^2 = \frac{1}{16}$. Thus, $(0,1)$ is equidistant from $(0,0)$ and $(1,0)$, forming an isosceles triangle. Hence, $(0,1) \notin \mathcal{T}$.

- **Admissible Set:** The only points of $AC$ that can coexist with all five points of $AB$ without creating a duplicate distance are $(0,2), (0,3)$, and $(0,4)$. Therefore, $\mathcal{T}$ must be a subset of the eight-point set $S$ defined in (8), i.e.,

$$\mathcal{T} \subseteq S.$$

**Impossibility of a Ninth Point**

Since $|S| = 8$, a set $\mathcal{T}$ of nine points must contain a point outside $S$. The remaining points of $\mathcal{L}$ are the interior points $(1,1), (1,2), (2,1)$ and the excluded boundary point $(0,1)$.

- If $(0,1)$ **is added**, we immediately form an isosceles triangle (as shown above).

- If an **interior point,** $(1,1)$**, is added**, then by (4),

$$|(1,1)(1,0)|^2 = |(1,1)(2,0)|^2 = \frac{1}{16}.$$

Thus, $(1,1)$ together with $(1,0)$ and $(2,0)$ forms an isosceles triangle, which contradicts the assumption that $\mathcal{T}$ is isosceles-free. The same phenomenon occurs for the other interior points.

Consequently, any ninth point forces the appearance of an isosceles triangle. No isosceles-free subset of $\mathcal{L}$ can have nine points; the maximal size is eight.

**Conclusion**

The largest possible cardinality of a subset of the fifteen lattice points that contains no three points forming an isosceles triangle is 8. Therefore, the smallest integer $n$ such that **every** choice of $n$ points necessarily contains an isosceles triangle is:

$$n = 8 + 1 = 9.$$

$$\boxed{9}$$

