# OpenReview forum: "LONG-HORIZON REASONING AGENT FOR OLYMPIAD- LEVEL MATHEMATICAL PROBLEM SOLVING"
_ICLR.cc/2026/Conference — Submitted to ICLR 2026_

### Official Review · Reviewer_bzvB · 2025-10-28

**Soundness:** 2
**Presentation:** 1
**Contribution:** 3
**Rating:** 4
**Confidence:** 3

**Summary:**

This paper introduces Intern-S1-MO, a long-horizon mathematical reasoning agent that aims to address the bottlenecks in large reasoning models, specifically the limitation of context length. During solving, the agent performs multi-round hierarchical reasoning that integrates reasoning, summarization, and verification, supported by a lemma-based memory management that effectively extends the reasoning depth. The paper further proposes OREAL-H with two critical components: hierarchical credit assignment and conjugate reward modeling to handle noisy process verification. In summary, Intern-S1-MO improves the reasoning precision and stability, which achieves state-of-the-art performance on Olympiad-level mathematics benchmarks.

**Strengths:**

1. The paper proposes an effective approach to overcome the context length limitation in large reasoning models.

2. Intern-S1-MO demonstrates strong overall performance on Olympiad-level mathematics benchmarks, surpassing previous SOTA.

**Weaknesses:**

1. Although the results are good, this paper is not well-written, and the presentation needs to be improved. Many details are missing in the current version, making it very hard to clearly understand.

2. The paper lacks sufficient description of the details of the proposed methods and training procedures. For example, the lemma-based memory management is a key contribution of the paper. However, the explanation of this method is not detailed enough, making it hard for readers to fully understand and reproduce it. Please refer to Questions for more details.

3. The information provided in the Appendix is incomplete and does not fully support the claims made in the main text.

4. It would be better to include some examples to help understand the process.

5. The proposed method requires several rounds to produce answers, while the other baseline seems to produce answers with only one inference. The experiments didn't mention the inference time, making the comparison unfair.

6. There is no code released during the review process. The authors only mention "Code and model will be released to benefit future research."

**Questions:**

1. The details of maintaining a structured lemma library are missing. During round $k$ of long-chain trajectories, how is a specific lemma chosen and explored from it? Also, how is an intermediate lemma decided to be updated in the memory system?

2. In Figure 2, it's unclear whether the input question in rounds 2 through $n-1$ is the same as the question in round 1, or if it includes partial solutions from previous rounds. Are the partial solutions stored in the Lemmas Library? How do the scores of the lemmas help in the reasoning process? Are they just references for the LLM, or do they have other roles? Also, the reason for keeping used lemmas in the library needs more explanation. Does the LLM solve the question from scratch in each round, and if so, does this library help the model skip steps that have already been solved?

3. In Figure 2, are the reasoner and summarizer the same model? On the right of the figure, there is a loop labeled "final solution draft," but without any explanation.

4. At line 413, is "Multi-Tune Reasoning" different from the "multi-round reasoning process" that includes memory management?

5. Is it also effective when applying the multi-round hierarchical reasoning framework (without finetuning) to other LLMs?

6. What is the critic $V$? Is it another LLM? What is the prompt for $V(s_t)$? How to train it?

7. What training dataset is used for the Verifier? Are the Lemma Verifier and Process Verifier the same models, or are they the same models with different prompts? How are they trained? Is a Process Verifier needed during inference?

8. In the section on "Conjugate Reward Modeling for Noisy Process Verification (PV)", it is mentioned that PV feedback is noisy. Does this mean that for a given lemma, PV can provide highly variable feedback for the lemma's correctness? Are there any examples of process verification? Are there any ablation studies conducted for the conjugate reward modeling? How effective is the conjugate reward model in denoising PV feedback?

9. In the Implementation section, which datasets (solution-based problems and proof-based problems) are used for RL? Do they include the problems used for evaluation? What is the cost of RL training for the LLM, including the number of questions in the training dataset and the number of fine-tuning iterations?

10. How many rounds are executed during the reasoning process? Is there a predefined limit, or does it depend on when the LLM performs the "commit answer" action?

11. What is the inference time for each model during the overall evaluation in Table 1?

12. Why is there no evaluation of IMO2025 in the ablation study in Table 2?



Typo and errors:

1. Some of the labels in the bars of Figure 1a are unclear.
2. At line 186, ProcessBench needs a citation.
3. At line 219, represent -> represents
4. At line 302, Appendix 5 is not provided.
5. At line 346, full implementation details, including scoring rubrics and problem filtering criteria, are not provided in Appendix 1.
6. In the appendix, at line 863, "Translated with DeepL.com (free version)" should be omitted.
7. In the appendix, Listing 1: LEMMA SEARCH, based on the content in the listing, it seems more like "Generating Solutions" than "LEMMA SEARCH".
8. In the appendix, Listing 2: Memory Management, it only describes how to extract lemmas from the output of LLM. It doesn’t include how the model chooses or utilizes the lemmas in the library.
9. "?)" citation error at line 442.
10. At page 10, there are duplicate references for "Intern-s1:...".

---

> ### Author Response · Authors · 2025-12-03
>
> **Response to Weakness1, 2, 3, 4 and Question1, 2, 3, 4 about the unclear presentation**
>
> We apologize for the ambiguity in the initial manuscript. We have updated the manuscript, adding a complete list of prompts used in the workflow in Appendix A, as well as some case studies in the Appendix F.
>
> **Response to Weakness5 about the inference cost**
>
> We have supplemented the details of the inference budget in our revised manuscript (Appendix B1), which shows the computational overhead we incurred to achieve the reported performance.
>
> **Response to Question5 about the generalization of the framework**
>
> Yes, our agentic system is universal.
>
> **Response to Question6, 7, and 8 about the verifier**
>
> We use the verifier agent as the critic. We have supplemented the details of it in our revised manuscript.
>
> **Response to Question9 and 10 about the implementation details**
>
> We have added implementation details in the revised manuscript (Appendix B), with B1 supplementing our system inference budget and B2 supplementing the details of the RL implementation.
>
> **Response to Question about the typo**
>
> Thanks for your suggestion! We have fixed the typos you mentioned in our revised manuscript.

---

### Official Review · Reviewer_ktR9 · 2025-10-29

**Soundness:** 3
**Presentation:** 3
**Contribution:** 3
**Rating:** 4
**Confidence:** 3

**Summary:**

This manuscript introduces Intern-S1-MO, a multi-agent framework designed to overcome the context length limitations that hinder large models on complex IMO-level math problems . The system operates via a multi-round loop of reasoning, summarization, and verification, storing intermediate proof steps as "lemmas" in a compact memory to break the constraints of single-pass inference. Trained using a novel reinforcement learning framework called OREAL-H, this agent achieved a silver-medal-equivalent score of 26 out of 35 on the non-geometry problems of IMO2025.
However, regarding the paper's two core claims ("overcoming context limitations" and "the multi-agent architecture"), the description of the key techniques is relatively vague, making it difficult to fully understand the author's specific methodological design.

**Strengths:**

The manuscript addresses one of the most challenging problems in the AI reasoning field: performing ultra-long-horizon reasoning on IMO-level tasks. This explicitly breaks through the bottleneck of current large reasoning models limited by context window size, representing a highly valuable research direction.

**Weaknesses:**

1. The paper describes Intern-S1-MO as a multi-agent system comprising a Reasoner, Summarizer, and Verifier. However, it is unclear whether these are three independently fine-tuned models or a single LRM playing three different roles via distinct prompts. If it is a single model, this raises a significant concern about self-confirmation bias. For example, would the model, when acting as the "Verifier," be biased toward favorably evaluating a proof it just generated as the "Reasoner"?

2. The update mechanism for the OREAL-H reinforcement learning gradient is underspecified. The RL reward (from the PV) is based on the Reasoner's final output. It is unclear if this gradient is only used to update the Reasoner's policy, or if it is also backpropagated to update the model's capabilities when performing the "Summarizer" and "Verifier" roles.

3.  The implementation of the "Theorem Verifier" (for intermediate lemmas) is vague. The paper states it uses "parallel sampling," which sounds like a self-consistency-based voting mechanism. However, the exact operational details are not provided. Furthermore, it is unclear why two different methods are necessary. If the trained PV is capable of "identifying the indices of steps containing logical fallacies," why is it not also used to verify the intermediate lemmas?

4.  The central claim of Intern-S1-MO is to overcome context limitations via its "Lemmas Libarary." However, the described process (Figure 2) involves feeding "Question + entire Lemmas Libarary" into the next reasoning round. This does not seem to solve the core problem. What happens when, after $n$ rounds of reasoning, the "Lemmas Libarary" itself grows to exceed the model's context window?

5.  The source and filtering criteria for the initial cold-start dataset (Section 3.2) are not specified, nor is the RL sampling strategy.
6.  The paper claims to use "512K tokens to solve a single problem," but this figure lacks crucial context. The authors do not specify the associated inference time, computational cost, or provide an efficiency comparison against single-pass, long-context models.

7.  The use of only 5 non-geometry problems for the IMO2025 benchmark represents a very small sample size. While achieving 26/35 points is an impressive result, any conclusions about SOTA performance drawn from such a limited sample have finite statistical robustness.

**Questions:**

See the weaknesses for details.

---

> ### Author Response · Authors · 2025-12-03
>
> **Response to Weakness1 about the unclear presentation of the workflow**
>
> > The paper describes Intern-S1-MO as a multi-agent system comprising a Reasoner, Summarizer, and Verifier. However, it is unclear whether these are three independently fine-tuned models or a single LRM playing three different roles via distinct prompts. If it is a single model, this raises a significant concern about self-confirmation bias. For example, would the model, when acting as the "Verifier," be biased toward favorably evaluating a proof it just generated as the "Reasoner"?
>
> Reasoner and summarizer are the same model, with different prompts (see Appendix 1) controlling their behavior. And the verifier is another model.
>
> **Response to Weakness2 about the unclear presentation of the RL**
>
> > The update mechanism for the OREAL-H reinforcement learning gradient is underspecified. The RL reward (from the PV) is based on the Reasoner's final output. It is unclear if this gradient is only used to update the Reasoner's policy, or if it is also backpropagated to update the model's capabilities when performing the "Summarizer" and "Verifier" roles.
>
> As mentioned in the previous reply, the reasoner and summarizer are the same model, so the parameters of this model will be updated during the RL process, while verifier will not be updated.
>
> **Response to Weakness3 about the unclear presentation of the verifier**
>
> > The implementation of the "Theorem Verifier" (for intermediate lemmas) is vague. The paper states it uses "parallel sampling," which sounds like a self-consistency-based voting mechanism. However, the exact operational details are not provided...
>
> In the lemma verification process, we use a self-consistency-based mechanism. Specifically, for each lemma, we use the theorem verifier to perform n parallel verifications, and the proportion of those correctly identified is used as the confidence score. We believe this improves the reliability of theorem verification, avoiding some false positives or false negatives. And we have supplemented the prompt for the lemma verifier in Appendix A.
>
> **Response to Weakness4 about the context windows**
>
> > The central claim of Intern-S1-MO is to overcome context limitations via its "Lemmas Libarary." However, the described process (Figure 2) involves feeding "Question + entire Lemmas Libarary" into the next reasoning round. This does not seem to solve the core problem. What happens when, after  rounds of reasoning, the "Lemmas Libarary" itself grows to exceed the model's context window?
>
> The Lemmas Library is a core setting in Intern-S1-MO, designed to compress long thought processes using lemmas. We sample some trajectories, and in practice, the average length ratio before and after compression can reach around 64:1.
>
> In addition, we set a maximum number of inference rounds, thus virtually eliminating the possibility of exceeding the context window.
>
> **Response to Weakness5 about the unclear presentation of the implementation details**
>
> > The source and filtering criteria for the initial cold-start dataset (Section 3.2) are not specified, nor is the RL sampling strategy.
>
> We have supplemented the implementation details in our revised manuscript (Section 4.1 and Appendix B).
>
> **Response to Weakness6 about the unclear presentation of token cost**
>
> > The paper claims to use "512K tokens to solve a single problem," but this figure lacks crucial context. The authors do not specify the associated inference time, computational cost, or provide an efficiency comparison against single-pass, long-context models.
>
> We have supplemented the details of the inference budget in our revised manuscript (Appendix B1), which shows the computational overhead we incurred to achieve the reported performance.
> It should be noted that 512k is a theoretical estimate based on our inference budget setting, which is the maximum output length per round multiplied by the maximum number of inference rounds (64k*8).
>
> **Response to Weakness7 about the robustness of the evaluation results**
>
> > The use of only 5 non-geometry problems for the IMO2025 benchmark represents a very small sample size. While achieving 26/35 points is an impressive result, any conclusions about SOTA performance drawn from such a limited sample have finite statistical robustness.
>
> IMO2025 is a highly challenging mathematical benchmark, a commonly used dataset for solving extremely difficult mathematical problems [1, 2].
>
> We acknowledge its limited sample size, so we also tested on several other benchmarks with larger problem sets, such as hmmt, aime, and cnmo. The results show that xxx also achieved state-of-the-art (SOTA) performance on these benchmarks.
>
> [1] Gemini 2.5 pro capable of winning gold at imo 2025
>
> [2] DeepSeekMath-V2: Towards Self-Verifiable Mathematical Reasoning

---

### Official Review · Reviewer_mMoy · 2025-11-02

**Soundness:** 2
**Presentation:** 1
**Contribution:** 3
**Rating:** 4
**Confidence:** 4

**Summary:**

While closed frontier labs have achieved great performance in olympiad-level competitions, the agentic workflow structure and training details underlying these successes remain underexplored in academia. Moreover, single-turn and solely prompt-based usage of frontier reasoning models falls far short of the reported success rates. This work therefore proposes open-source agentic workflows and RL training strategies for solving olympiad-level math competitions.

The proposed agentic workflow aims to decompose complex reasoning into a scaffold of multiple agents. First, hierarchical reasoning decomposition is performed through multiple rounds of maintaining and updating a lemma database, leveraging a solver stage (for reasoning), a summarization stage that converts the reasoning trace into structured lemma memory, and a lemma verification stage. Next, another agent writes the complete solution given the refined lemma database. Finally, the complete solution undergoes multiple rounds of verification loops through interaction with a verifier agent to produce the final solution.

Further, the authors propose a new RL framework called OReal-H, which is claimed to improve the workflow's performance. The resulting solution, called Intern-S1-MO, is shown to outperform all single-turn performances of frontier reasoning models. Furthermore, the paper argues that even the distilled 8B model derived from Intern-S1-MO, called Intern-S1-mini-MO, performs on par with these frontier reasoning models and even outperforms all of them on IMO 2025.

**Strengths:**

1. The reported performance of the proposed solution (Intern-S1-MO) is strong, especially its performance on IMO, where it achieves a score of 26/35 (excluding problem 2 of the exam, which was a geometry problem), while the best single-turn frontier model achieved a score of 14.
2. Distilling an 8B model from Intern-S1-MO is an impactful contribution to the open-source community, as surprisingly, when the proposed workflow is equipped with this relatively small model, it achieves a performance of 17/35, still outperforming the best single-turn frontier model.
3. The idea of tracking past reasoning exploration explicitly with a structured lemma database and refining this database through multiple rounds represents clever and novel decisions in agentic workflow design for math agents.

**Weaknesses:**

1. **Clarity of the proposed agentic workflow could be improved:** The prompts provided in Appendix A, while helpful, are not sufficient to fully understand how each component of the workflow operates. The prompts used for the verifier component shown in Figure 2 are not provided across all stages. Additionally, it is unclear what prompt is used for obtaining the "long-chain trajectory" as it appears in the middle column of Figure 2 (rounds 2,...,n-1). Including all these prompts would be valuable for better understanding the proposed workflow's soundness. Regarding lemma search (line 158), the paper mentions "refining the model via prompt engineering and targeted training, explicitly enabling it to produce partial deductive progress in single-turn attempts," but further explanation of how this targeted training is conducted and how the data is curated for this step would strengthen the paper. Similarly, for process verification (lines 183-185), while the paper describes training "a specialized process verifier using synthetic cold start data with outcome supervision" and employing DPO, additional details about the training process, data curation, and the advantages of this approach would be beneficial for readers to fully appreciate the contribution.

2. **The proposed RL framework would benefit from additional clarification:** While the paper attempts to provide theoretical understanding of the RL framework, certain aspects of the derivation could be clearer. Specifically, π_φ is not defined explicitly, and assuming it represents the high-level policy, the rigorous transition from Equation 1 to Equation 3 could be better explained. The explanation of the dedicated critic V^H(s_t) would benefit from more details on its practical estimation and implementation. Including pseudocode with high-level abstraction of the RL implementation and a final simplified RL loss would help readers better assess the framework's soundness. Additionally, more discussion on data curation across different stages of RL training would be valuable.

3. **Cost-performance profiling and comparison with simpler baselines:** A comparison with a simple baseline of solver and verifier agents in a loop would provide important context, as [1] demonstrated that such a baseline can achieve 35/35 on IMO 2025. Since the proposed workflow in Figure 2 already contains this component in "round n," this ablation would help clarify how this component performs standalone. Understanding the computational cost of running only "round n" compared to the full Intern-S1-MO workflow at comparable IMO performance levels would provide valuable insights into the efficiency gains of the proposed approach.

4. **Evaluation details require expansion:** The implementation details and evaluation rubrics are currently missing, with the paper referencing Appendix 1 (line 346), which appears to be absent. Including these details would strengthen the reproducibility of the work.

5. **Ablation studies could be more clearly presented:** The meaning of "Single-turn with Agents" requires clarification. Additionally, explaining how complexity can be added incrementally in an agentic workflow context would help readers better understand the incremental benefits of each component. These clarifications would be valuable to address during the rebuttal phase.

References:
[1] Huang, Yichen, and Lin F. Yang. "Gemini 2.5 pro capable of winning gold at imo 2025." arXiv preprint arXiv:2507.15855 7 (2025).

**Questions:**

Questions:
1. Could you please elaborate on what each model in Table 2 represents? Specifically, what is the exact workflow structure for each of these models?
2. Could you provide more details on how the verifier is trained in the final round of the workflow? How is the data curated for this training? I have similar questions regarding the "targeted training" mentioned in line 159.
3. Could you provide all the prompts used in the agentic framework shown in Figure 2 across all rounds and components (agents)? This would greatly improve the clarity and reproducibility of the agentic framework.
4. What is the high-level pseudocode for the final RL implementation, and what is the final loss function used? This would help clarify the proposed OReal-H framework.

---

> ### Author Response · Authors · 2025-12-03
>
> **Response to Weakness1, Question2 and Question3 about the unclear presentation of the workflow**
>
> > Clarity of the proposed agentic workflow could be improved: The prompts provided in Appendix A, while helpful, are not sufficient to fully understand how each component of the workflow operates... Similarly, for process verification (lines 183-185), while the paper describes training "a specialized process verifier using synthetic cold start data with outcome supervision" and employing DPO, additional details about the training process, data curation, and the advantages of this approach would be beneficial for readers to fully appreciate the contribution.
>
> Thanks for your suggestion! We have made some revisions to the manuscript based on it.
> Specifically, we have supplemented all the prompts used throughout the workflow in Appendix A, covering each stage described in Figure 2. And we have added the details of the process verifier in Section 2.
>
> **Response to Weakness2 and Question4 about the unclear presentation of RL**
>
> > The proposed RL framework would benefit from additional clarification: While the paper attempts to provide theoretical understanding of the RL framework, certain aspects of the derivation could be clearer. Specifically, π_φ is not defined explicitly, and assuming it represents the high-level policy, the rigorous transition from Equation 1 to Equation 3 could be better explained. The explanation of the dedicated critic V^H(s_t) would benefit from more details on its practical estimation and implementation. Including pseudocode with high-level abstraction of the RL implementation and a final simplified RL loss would help readers better assess the framework's soundness. Additionally, more discussion on data curation across different stages of RL training would be valuable.
>
> In our revised manuscript, we have clarified the corresponding parts in RL framework in Section 3 and  supplemented  a pseudocode with high-level abstraction of the RL implementation in Appendix C.
>
> **Response to Weakness3 about the comparison with simpler baselines**
>
> > Cost-performance profiling and comparison with simpler baselines: A comparison with a simple baseline of solver and verifier agents in a loop would provide important context, as [1] demonstrated that such a baseline can achieve 35/35 on IMO 2025. Since the proposed workflow in Figure 2 already contains this component in "round n," this ablation would help clarify how this component performs standalone. Understanding the computational cost of running only "round n" compared to the full Intern-S1-MO workflow at comparable IMO performance levels would provide valuable insights into the efficiency gains of the proposed approach.
>
> To clarify, we have made a comparison with the simpler baseline in the ablation study in Section 4.3, namely the first and last rows of Table 2. The results show that performing only single-turn simple inference results in a significant performance drop, which is why we advocate for multi-turn inference to improve performance by extending the context length.
>
> As for the work [1], its evaluation logic is not rigorous. There are some discussions on twitter about their real examination results under human experts' judges.
>
> In addition, we have supplemented the details of the inference budget in our revised manuscript (Appendix B1), which shows the computational overhead we incurred to achieve the reported performance.
>
> [1] Huang, Yichen, and Lin F. Yang. "Gemini 2.5 pro capable of winning gold at imo 2025." arXiv preprint arXiv:2507.15855 7 (2025).
>
> **Response to Weakness4 about the unclear presentation of evaluation details**
>
> > Evaluation details require expansion: The implementation details and evaluation rubrics are currently missing, with the paper referencing Appendix 1 (line 346), which appears to be absent. Including these details would strengthen the reproducibility of the work.
>
> We have supplemented the details of evaluation system in our revised manuscript (Appendix D).
>
> **Response to Weakness5 and Question1 about the unclear presentation of ablation study**
>
> > Ablation studies could be more clearly presented: The meaning of "Single-turn with Agents" requires clarification. Additionally, explaining how complexity can be added incrementally in an agentic workflow context would help readers better understand the incremental benefits of each component. These clarifications would be valuable to address during the rebuttal phase.
>
> We have added more description in the caption of Table 2 to help clarify the component in our framework. And we have described how we add components to the Agentic workflow step by step in our revised manuscript (Section 4.3).

---

### Official Review · Reviewer_taYP · 2025-11-11

**Soundness:** 3
**Presentation:** 3
**Contribution:** 3
**Rating:** 4
**Confidence:** 3

**Summary:**

Authors proposes a lemma-memory, multi-round agent with distinct Reasoner/Summarizer/Verifier roles to push beyond single‑context limits. Also authors Introduces OREAL‑H: hierarchical credit assignment + a conjugate (Beta‑Bernoulli) reward for noisy process verification The results are strong with reports state-of-the-art scores on several math benchmarks and IMO non‑geometry 26/35 with up to ~512K tokens/problem.

**Strengths:**

The paper is well-motivated, tackling the important and clearly defined challenge of long-horizon reasoning. Its technical contributions are presented through a clear agent architecture and are supported by a broad, rigorous empirical evaluation. This evaluation includes thorough ablations that definitively quantify the contribution of each component, making the paper's methodological strengths both easy to grasp and convincingly validated.

**Weaknesses:**

The empirical evaluation, though extensive, suffers from a lack of clarity regarding the primary performance metric. The paper states it uses a "pass@1" score, defined as the "expected score from the best single attempt," but derives this from 16 rollouts per problem. This methodology effectively reports a "best-of-16" result, which artificially inflates performance compared to a true single-sample evaluation and makes it difficult to compare against baselines that may not have been evaluated with an equivalent sampling strategy. A clearer justification for this protocol and a demonstration that all baselines were compared under identical conditions is necessary. Furthermore, the claim of superior performance is difficult to fully assess without a discussion of computational fairness. The agent's architecture, which consumes approximately ~512K tokens/problem through parallel verification and multi-round reasoning, represents a significant computational budget. It is unclear whether the baseline models, which are predominantly single-pass, were allocated a comparable "thinking budget" in terms of total tokens, rollouts, or verification calls. A comparison normalized for computational cost would provide a more rigorous foundation for the performance claims.

**Questions:**

*Q1*. The paper argues that lemmas extend reasoning beyond context limits. Could you illustrate this with a concrete example from your experiments? For instance, describe a problem where the agent discovered a non-obvious intermediate result like a specific invariant or inequality that was not in the initial reasoning trace?

*Q2*. Can authors explain how storing any single lemma fundamentally altered the strategic possibilities in the next round?

*Q3*. A qualitative comparison of proof strategies generated under the conjugate reward versus a hypothetical raw k/n reward would be very insightful for understanding method practical effect. What specific, observable changes in the agent's proof-writing style did this training incentive produce? For example, did you see a trend towards more concise and robust arguments?

*Q4*. The system involves a trade-off between exploration, summarization, and verification. Was there a heuristic or a learned halting policy to decide when further exploration was unlikely to be fruitful and it was time to commit to a final answer?

---

> ### Author Response · Authors · 2025-12-03
>
> **Response to Weakness1 about evaluation metrics**
>
> > The empirical evaluation, though extensive, suffers from a lack of clarity regarding the primary performance metric. The paper states it uses a "pass@1" score, defined as the "expected score from the best single attempt," but derives this from 16 rollouts per problem. This methodology effectively reports a "best-of-16" result, which artificially inflates performance compared to a true single-sample evaluation and makes it difficult to compare against baselines that may not have been evaluated with an equivalent sampling strategy. A clearer justification for this protocol and a demonstration that all baselines were compared under identical conditions is necessary.
>
> We apologize for the typo here. To clarify, we have not used "best-of-n" as the evaluation metric. We follow the setting in [1] and use the unbiased pass@1 as the evaluation metric, with the formula: Pass@k = 1 - C(n-c, k) / C(n, k), where c denotes the number of correct samples and n denotes the total number of samples.
>
> We have corrected the corresponding statement in our revised manuscript (Section 4.1).
>
> [1] Chen M. Evaluating large language models trained on code[J]. arXiv preprint arXiv:2107.03374, 2021.
>
> **Response to Weakness2 about computational cost**
>
> > Furthermore, the claim of superior performance is difficult to fully assess without a discussion of computational fairness. The agent's architecture, which consumes approximately ~512K tokens/problem through parallel verification and multi-round reasoning, represents a significant computational budget. It is unclear whether the baseline models, which are predominantly single-pass, were allocated a comparable "thinking budget" in terms of total tokens, rollouts, or verification calls. A comparison normalized for computational cost would provide a more rigorous foundation for the performance claims.
>
> For baseline models, we default to an inference length budget of 64k.
> Current models, limited by their architecture and inference infrastructure, cannot achieve an ultra-long output of 512k in a single round. We believe this inherent constraint hinders their potential for solving highly complex, long-context problems.
> Consequently, our core contribution is the development of a stable and effective scheme for extending the practical inference length. We demonstrate that this length extension leads to substantial performance enhancements.
> In addition, we have supplemented the details of the inference budget in our revised manuscript (Appendix B1), which shows the computational overhead we incurred to achieve the reported performance.
>
> **Response to Question1 and Question2 about the influence of lemmas**
>
> > Q1. The paper argues that lemmas extend reasoning beyond context limits. Could you illustrate this with a concrete example from your experiments? For instance, describe a problem where the agent discovered a non-obvious intermediate result like a specific invariant or inequality that was not in the initial reasoning trace?
>
> > Q2. Can authors explain how storing any single lemma fundamentally altered the strategic possibilities in the next round?
>
> Thanks for your suggestion! We have added a case study in our revised manuscript (Appendix F), which demonstrates the impact of intermediate lemmas on the reasoning process.
>
> **Response to Question3 about the influence of RL**
>
> > Q3. A qualitative comparison of proof strategies generated under the conjugate reward versus a hypothetical raw k/n reward would be very insightful for understanding method practical effect. What specific, observable changes in the agent's proof-writing style did this training incentive produce? For example, did you see a trend towards more concise and robust arguments?
>
> Thanks for your suggestion! We have added a case study in our revised manuscript (Appendix F), which compares the differences in model output style before and after RL and demonstrates that our RL solution is effective.
>
> **Response to Question4 about the indicator of reasoning stop**
>
> > Q4. The system involves a trade-off between exploration, summarization, and verification. Was there a heuristic or a learned halting policy to decide when further exploration was unlikely to be fruitful and it was time to commit to a final answer?
>
> We employ several conditions to limit the model's inference budget. Firstly, as mentioned in Appendix A.1, the reasoner's system prompt requires it to output the keyword "I have found a complete solution" while finding the final solution. And once this keyword is detected, no further exploration will proceed. Secondly, we limit the maximum number of inference rounds. As mentioned in Appendix B1, in practice, we limit the maximum number of inference rounds to 8.

---

### Meta-Review · Area_Chair_H4dd · 2026-01-06

**Summary:**

The final decision is to reject this submission. While the consensus among reviewers initially highlighted the paper's strong reported results on IMO, HMMT, and AIME benchmarks, alongside a clear and novel agent architecture utilizing a lemma database, significant barriers prevented acceptance. The primary collective concern that dictated this outcome was the ambiguity regarding fair evaluation metrics and computational cost normalization. Specifically, all reviewers questioned the fairness of comparing the proposed method, which utilizes up to 512K tokens per problem, against standard baselines without a normalized assessment of inference time or token budget. Despite the evident promise of the approach in overcoming context-window limits, the lack of rigorous cost-controlled comparisons and reproducibility artifacts ultimately weighed heavily against the submission.

**Reviewer Concerns:**

The authors provided a comprehensive rebuttal that successfully addressed several methodological ambiguities and clarity issues. On the side of completeness, the revision effectively clarified the use of unbiased pass@k metrics, bringing necessary rigor to the evaluation framework. Furthermore, the inclusion of detailed prompts, case studies, and specific implementation details regarding the 'Verifier' and 'Reasoner' roles significantly improved the manuscript's readability and transparency. The addition of broader benchmarks helped mitigate concerns regarding the small sample size of the IMO dataset.

However, the soundness of the evaluation regarding computational efficiency remains a critical, outstanding barrier. While the authors added inference budget details, they failed to provide a normalized comparison that accounts for the massive disparity in compute usage between their agentic workflow and the baselines. The core dialectical conflict, i.e., high performance is partly a function of massive inference scaling rather than purely architectural innovation, was not sufficiently resolved. Additionally, while implementation details were expanded, the absence of released code leaves reproducibility concerns lingering, particularly for Reviewers mMoy and bzvB, hindering the ability of the community to verify the complex multi-round interactions described.

**Reviewer Scores:**

Based on the rebuttal's effectiveness, the reviewers' scores are projected to diverge. Reviewer taYP and Reviewer ktR9 would likely increase their scores from 4 to 6 (borderline accept). For these reviewers, the clarification of the pass@k metric, the distinct roles of the agent components, and the logic regarding lemma memory compression were sufficient to outweigh the remaining concerns about compute normalization. They viewed the methodological clarifications as having cleared the primary hurdles to understanding the contribution.

Conversely, Reviewer mMoy and Reviewer bzvB are expected to maintain their scores at 4 (borderline reject). For Reviewer mMoy, although clarity was improved, the 'light' handling of compute normalization and the absence of a rigorous cost-performance analysis meant the primary barrier was not dismantled. Similarly, Reviewer bzvB would likely remain unconvinced due to the combination of weak fairness comparisons and the lack of code release, which are essential for validating such a complex, high-compute system. Consequently, the weighted consensus tilts toward rejection due to these persistent issues regarding fairness and reproducibility.

---

### Decision · Program_Chairs · 2026-01-26

Reject